

**Title:**

**Carbon budget assessment of an irrigated wheat and maize rotation cropland with high**

**groundwater table in the North China Plain**

**Quan Zhang[1,2], Hui-Min Lei[2], Da-Wen Yang[2], Lihua Xiong[1], Beijing Fang[2]**

[1]State Key Laboratory of Water Resources and Hydropower Engineering Science, Wuhan

University, Wuhan, China

[2]State Key Laboratory of Hydroscience and Engineering, Department of Hydraulic

Engineering, Tsinghua University, Beijing, China

Correspondence to:

Q. Zhang (quan.zhang@whu.edu.cn) and H. M. Lei (leihm@tsinghua.edu.cn)

Tel: 86-(0)10-6278-3383

Fax: 86-(0)10-6279-6971



**Abstract:**
Carbon sequestration of cropland has the potential to mitigate global greenhouse gas
emissions. To understand such sequestration of an irrigated wheat-maize rotation cropland
with high groundwater table in the North China Plain, the carbon budget and its components
are estimated with a comprehensive field experiment by combining eddy covariance
technique, soil respiration experiment differentiating heterotrophic and below-ground
autotrophic respirations, and biometric measurements in a relatively wet year from October
2010 to October 2011. In the experimental period of a whole winter-wheat and summer-maize
cycle, the Net Ecosystem Exchange, Gross Primary Productivity, Ecosystem Respiration, soil
heterotrophic respiration, below-ground autotrophic respiration and above-ground autotrophic
respiration are $-437.9$, $1078.2$, $640.4$, $376.8$, $135.5$ and $128.0$ gC m$^{-2}$, respectively for wheat
season, and are $-238.8$, $779.7$, $540.8$, $292.2$, $115.4$ and $133.2$ gC m$^{-2}$, respectively for maize
season. The experiment allows for estimations of Net Primary Productivity, Net Ecosystem
Productivity and Net Biome Productivity. The Net Biome Productivity are $58.8$ and $3.9$ gC m$^{-}$
$^2$ for wheat and maize season, indicating that wheat is a carbon sink and maize is close to
carbon neutral. However, compensated by the net ecosystem carbon release in two rotation
periods, Net Biome Productivity of the whole wheat-maize rotation cycle is $12.8$ gC m$^{-2}$ yr$^{-1}$
in the experimental year, indicating this cropland remains a weak carbon sink under the
specific climatic conditions and field conditions with a high groundwater table. The cropland
has a higher ecosystem carbon use efficiency (CUE) than other terrestrial ecosystems,
indicating that the agro-ecosystem is more efficient in harvesting $CO_2$ from the atmosphere.
This irrigated wheat-maize rotation cropland with high groundwater table has higher CUE



than other croplands, implying that the cropland management of full irrigation and
fertilization promotes carbon accumulation in crops.
**Key words:** Carbon; Irrigation; Wheat; Maize; Agro-ecosystem; North China Plain;



## Introduction

Terrestrial ecosystem carbon research has been attracting growing interest in the context of
climate change (Falkowski et al., 2000; Poulter et al., 2014; Forkel et al., 2016), and the
continuous Net Ecosystem Exchange (NEE) observations using eddy covariance have
dramatically fostered our understanding of terrestrial ecosystem carbon budget (Aubinet et al.,
2000; Baldocchi et al., 2001; Falge et al., 2002b). However, the eddy covariance technique
only measures the net ecosystem exchange with the atmosphere (i.e., the NEE of $CO_2$).
Though NEE can be partitioned into Gross Primary Productivity (GPP) and Ecosystem
Respiration (ER) using appropriate algorithms (Falge et al., 2002a; Reichstein et al., 2005), it
remains insufficient to fully unravel the mechanisms that control terrestrial ecosystem carbon
budget, because the detailed carbon budget components remain lacking for most typical
ecosystems.
The prevailing large-scale carbon budget evaluations largely depend on numerical models
(e.g., Piao et al., 2012; Chen et al., 2016; Thompson et al., 2016) that are generally difficult to
calibrate due to lacking measured carbon budget components, which consist of carbon
assimilation (i.e., GPP), carbon release in the forms of soil heterotrophic respiration ($R_H$),
above-ground autotrophic respiration ($R_{AA}$) and below-ground autotrophic respiration ($R_{AB}$).
These different carbon components contain different biological and biophysical processes
(Moureaux et al., 2008) that may respond differently to climatic variables, environmental
factors and ecosystem management practices (Ekblad et al., 2005; Zhang et al., 2013).
Differentiating these carbon budget components is therefore required both to calibrate models



diagnosing the carbon processes and, to understand the response of terrestrial ecosystem
carbon balance to changing climatic and environmental conditions (Heimann and Reichstein,
2008). Additionally, obtaining the different carbon components remains a prerequisite to
identify which components are critical in determining whether an ecosystem is a carbon sink
or source. However, the most recent efforts on detailed carbon budget components remain
only limited to a few studies on forests (Iglesias et al., 2013; Wu et al., 2013) and agro-
ecosystems (e.g., Moureaux et al., 2008; Wang et al., 2015; Demyan et al., 2016).
Among all the terrestrial ecosystems, the agro-ecosystems play an important role in regulating
the global carbon balance (Lal, 2001; Bondeau et al., 2007; Özdoğan et al., 2011; Taylor et al.,
2013) and, have great potentials in mitigating the global carbon emission through cropland
management (Sauerbeck, 2001; Freibauer et al., 2004; Smith, 2004; Hutchinson et al., 2007;
van Wesemael et al., 2010; Ciais et al., 2011; Schmidt et al., 2012). Previous studies
suggested that management practices (e.g., irrigation, fertilization and residue removal, etc.)
significantly impact the cropland carbon budget (Baker and Griffis, 2005; Béziat et al., 2009;
Ceschia et al., 2010; Eugster et al., 2010; Drewniak et al., 2015; de la Motte et al., 2016; Hunt
et al., 2016; Vick et al., 2016). However, it remains difficult to identify the key factors
determining cropland carbon behaviors because of the relatively limited field observations
(Kutsch et al., 2010), prompting the interest of comprehensive carbon budget assessments
across cropland management types.
Over the past two decades, agro-ecosystem carbon studies have mainly focused on variations
in the integrated ecosystem exchange with atmosphere (i.e., NEE) or its two derived



components GPP and ER using eddy covariance, such as for wheat (Gilmanov et al., 2003;
Anthoni et al., 2004a; Moureaux et al., 2008; Vick et al., 2016), maize (Verma et al., 2005),
sugar beet (Aubinet et al., 2000; Moureaux et al., 2006), potato (Anthoni et al., 2004b;
Fleisher et al., 2008), soybean-maize rotation cropland (Gilmanov et al., 2003; Hollinger et
al., 2005; Verma et al., 2005; Grant et al., 2007), and winter wheat-summer maize cropland
(Zhang et al., 2008; Lei and Yang, 2010). But the eddy covariance technique alone cannot
capture the agro-ecosystem lateral carbon fluxes associated with harvesting, residue treatment
and manure addition, which significantly influence carbon budget (Kutsch et al., 2010). To
overcome this problem, some studies investigated Net Biome Productivity (NBP) using the
eddy covariance technique complemented with ancillary carbon measurements (i.e., harvest,
residue, manure etc.) (Kutsch et al., 2010). But only a few studies have reported detailed
carbon budget components (e.g., Moureaux et al., 2008; Aubinet et al., 2009; Jans et al., 2010;
Wang et al., 2015; Demyan et al., 2016). More importantly, there remains no consensus on
whether agro-ecosystem is a carbon sink or source. To satisfy the increasing need of
understanding agro-ecosystem carbon behaviors, comprehensive field carbon budget
evaluations remain imperative.
For one of the most important food production regions in China - the North China Plain,
which provides more than 50 % wheat and 33 % maize to the whole nation (Kendy et al.,
2003) and, therefore guarantees the national food security. Irrigation is very common in North
China Plain because of the frequent spring drought. There are two major types of irrigation
method in the North China Plain, one is pumping water from groundwater, leading dramatic





groundwater table decline (more than 20 m) on the piedmont plain of Mount Taihang, the
other is withdrawing water from the Yellow River, resulting in very high groundwater table
(less than 5 m) along the Yellow River (Cao et al., 2016) (See Fig. 1). Such high groundwater
table along the Yellow River is the major different feature from the groundwater-fed cropland
(Shen et al., 2013). Wang et al. (2015) suggested that the groundwater-fed cropland
(Luancheng site in Fig. 1) of the piedmont plain of Mount Taihang is losing carbon at the rate
of 77 gC m$^{-2}$ yr$^{-1}$. Another two studies also reported that the cropland along the Yellow River
was a carbon source with the groundwater table fluctuating between 0.3 to 5.0 m (Li et al.,
2006; Luo et al., 2008; Lei and Yang, 2010). But it remains unknown whether such conclusion
holds across the whole North China Plain region with diverse field microclimates and
management practices. Lacking such knowledge limits our comprehensive understanding of
how this cropland contributes to regional and global carbon cycle, also motivating this study.
In light of this, we conducted a field experiment in an irrigated wheat-maize rotation cropland
with abnormally high groundwater table (i.e., water logging happened). This study provides a
comprehensive carbon budget assessment by combining the eddy covariance, soil respiration
and biometric measurements for a whole wheat-maize cycle from October 2010 to October
2011. These measurements (1) allow investigating the seasonal variations in the integrated
flux NEE, GPP and ER; (2) provide the three components of ER, i.e., $R_H$, $R_{AB}$ and $R_{AA}$; and
(3) further allow estimations of Net Primary Productivity (NPP), Net Ecosystem Productivity
(NEP) and Net Biome Productivity (NBP) with two independent methods.



**Materials and methods**
*Site description and field management practice*
The experiment is conducted in a rectangular-shaped (460 m × 280 m) field, which is located
in a typical flat cropland (36° 39' N, 116° 03' E, Weishan site in Fig. 1) in the North China
Plain. The soil is silt loam with the field capacity and saturation point of 0.33 and 0.45 $m^3$ $m^{-3}$
for the upper 5 cm soil. The mean annual precipitation and mean air temperature were
532 mm and +13.3 °C from 1984 to 2007. The double cropping system of winter wheat and
summer maize is the typical tillage style. Winter wheat is generally sown at the start of
October and is harvested in the middle of June in the following year, and the residue is left to
the ground without tillage at harvest. Summer maize is generally sown following wheat
harvest and, is harvested in October. Prior to sowing wheat for next season, a thorough tillage
is conducted to fully smash maize residue and blend the residue powder with the ~20 cm
surface soil. Nitrogen fertilizer is commonly applied in North China Plain, the field inventory
of Weishan site shows the nitrogen application are 35 gN $m^{-2}$ in the wheat season and
20 gN $m^{-2}$ in the maize season in the experimental period.
(Fig. 1 here)
Wheat is irrigated by water withdrawal from the Yellow River with an irrigation of about 150
mm. Maize is rarely irrigated because precipitation is generally sufficient in maize season.
Such irrigation method by withdrawing water from the Yellow River, causes a very high
groundwater table (generally fluctuates between 0 and 4 m) in this region (see Fig. 1). Field
water logging casually appears in maize season because of the high precipitation together



with the preceding high groundwater table, contributing to quite a special humid microclimate
and saturated soil condition that rarely appear in upland cropland.
During the experimental period, winter wheat was sown on October 23$^{rd}$, 2010 with the plant
density of 775 plants m$^{-2}$ and a ridge spacing of 0.26 m, and was harvested on June 10$^{th}$, 2011;
Summer maize was sown on June 23$^{rd}$, 2011 with the plant density of 4.9 plants m$^{-2}$ and a
ridge spacing of 0.63 m, and was harvested on Sep. 30$^{th}$, 2011; The next wheat season started
from October 11$^{th}$, 2011, the period from October 23$^{rd}$, 2010 through October 10$^{th}$, 2011 is
selected as the study period for annual carbon budget evaluations. Evaluating with
precipitation measurements from 1953 to 2012 by China Meteorological Administration
(http://data.cma.cn/), the main growing season of wheat (March, April and May in 2011) was
near 'normal' with estimated 3-month Standard Precipitation Index (SPI3) of −0.31, while the
main growing season of maize (July, August and September in 2011) was 'moderately wet'
with SPI3 of 1.16 (World Meteorological Organization, 2012). Water logging happened in late
August and early September during the experimental period, resulting from the full irrigation
of wheat season, high precipitation of maize season, and a high preceding groundwater table
associated with the high summer precipitation in 2010 (SPI3 of July, August and September
was 1.51 labelled as 'very wet'). The specially wet field condition and the associated
microclimate significantly distinguished the experimental period from other years.
***Environmental measurements***
The meteorological variables are measured continuously at 30 min interval with a standard
meteorological station. Among these variables are the air temperature ($T_a$) and relative



humidity (RH) (HMP45C, Vaisala Inc., Helsinki, Finland) at the height of 1.6 m, precipitation
($P$) (TE525MM, Campbell Scientific Instruments Inc., Logan, UT, USA) and photosynthetic
photon flux density (PPFD) (LI-190SA, LI-COR Inc., Lincoln, NE, USA) at 3.7 m above the
ground. The 30 min interval edaphic measurements include soil temperature ($T_S$) (109-L,
Campbell Scientific Instruments Inc.), volumetric soil moisture ($\theta$) (CS616-L, Campbell
Scientific Instruments Inc.) and soil matric potential ($\psi$) (257-L, Campbell Scientific
Instruments Inc.) at the depth of 5 cm. The groundwater table (WT) (CS420-L, Campbell
Scientific Instruments Inc.) is also measured close to the meteorological station at 30 min
interval.
***Biometric measurements***
To trace crop development and carbon storage, the canopy height ($H_C$), Leaf Area Index
(LAI), crop Dry Matter (DM), and carbon content of crop organs are measured at an interval
of 7-10 days. The inclement weather and water logging conditions, however, occasionally
forced the measuring interval to two weeks or even longer. The $H_C$ and LAI are measured
with a ruler and LAI-2000 (LI-COR Inc.) at 10 points randomly distributing in the field.
When measuring DM, 4 points are selected randomly at the start of growing season, plant
samples are then collected at these 4 points across the experimental period. At each point, 10
plant samples are collected in the wheat season, and 3 plant samples are collected during the
maize season. To reduce sampling uncertainty at harvest, 200 plants and 5 plants at each point
are collected during the wheat season and maize season, respectively. The crop organs are
separated and oven-dried at 105 °C for kill-enzyme torrefaction for half an hour, and finally



oven-dried at 75 °C until constant weight. The crop samplings together with crop density
allow estimations of field biomass (Dry Matter). The carbon content is analyzed by
combustion oxidation-titration method (National Standards of Environmental Protection of
the People's Republic of China, 2013).
*Eddy covariance measurements*
The eddy covariance system consists of an infrared gas analyzer (LI-7500, LI-COR Inc.) and
a three dimensional sonic anemometer (CSAT3, Campbell Scientific Instruments Inc.) that are
mounted 3.7 m above the ground. The post processing includes NEE calculation, quality
control (Mauder and Foken, 2004) and gap filling of missing measurements during either the
rain event or the nighttime when the atmosphere is stable. In gap filling procedure, small gaps
within 2 hours are filled using linear regression, while other gaps are filled using Mean
Diurnal Variation (MDV) method (Falge et al., 2001). NEE is further partitioned to derive
GPP and ER (Reichstein et al., 2005; Lei and Yang, 2010) by assuming diurnal and nocturnal
respirations share the same temperature response, the temperature response function of
respiration (Eq. (1)) is first fitted with nocturnal carbon flux and temperature, diurnal
respiration is then extrapolated by using the fitted nocturnal temperature relationship as,
$\mathrm{ER} = \mathrm{ER_{ref}} \exp(bT_S)$ ,                    (1)
where $\mathrm{ER_{ref}}$ is the reference respiration, i.e., respiration at 0 °C, and $b$ is the temperature
sensitivity parameter that is associated with the commonly used temperature sensitivity
coefficient $Q_{10}$ via,
$Q_{10} = \exp(10b)$ .                    (2)





Note that the eddy covariance system failed from October 23$^{rd}$, 2010 to April 1$^{st}$, 2011 in the
wheat season, Support Vector Regression (SVR) method is then used to calculate GPP and ER
(Cristianini and Shave-Taylor, 2000), NEE is finally derived as the difference between GPP
and ER (see Appendix A for the details).
***Soil respiration measurements and synthesis***
Soil respiration was measured between 13:00 and 15:00 every day from April through
September of 2011, except for days with rain events and field water logging conditions, using
a portable soil respiration system LI-8100 (LI-COR Inc.). The below-ground autotrophic
respiration ($R_{AB}$) and heterotrophic respiration ($R_H$) are differentiated using the root exclusion
method (Wan and Luo, 2003; Jassal et al., 2012; Zhang et al., 2013), and these measurements
allow estimating the $R_{AB}$ contribution ratio to $R_S$ (Zhang et al., 2013). The heterotrophic
respiration is measurement of treatment without root, total soil respiration is the measurement
of treatment with root, and the difference gives the below-ground autotrophic respiration. To
reduce the uncertainty associated with spatial variability, we set three replicated pairs of
comparative treatments (i.e., with root and without root). To assess the seasonal variations and
total amount of soil respiration, the seasonal continuous $R_H$ record is then calculated using the
$Q_{10}$ model by incorporating soil moisture as follows:
$$R_H = A \exp(B T_s) \cdot f(\theta) , \tag{4}$$
$$f(\theta) = \begin{cases} 1, & \theta \leq \theta_f \\ a(\theta - \theta_f)^2 + 1, & \theta > \theta_f \end{cases} , \tag{5}$$
where $\theta_f$ is the field capacity. The other parameters are inferred from the $R_H$ measurements,




where A=1.16, B=0.0503 and $a$= −44.9 (Zhang et al., 2013).
The $R_{AB}$ of wheat is assumed to be 0 before March 14 due to the low plant biomass, while $R_{AB}$
of other period is estimated based on $R_H$ record and the contribution ratio of the $R_{AB}$ to $R_S$
(Zhang et al., 2013). The seasonal continuous contribution ratio of $R_{AB}$ is inferred from the
daily single point measurement using the linear interpolation (Fig. 2), such estimation is
reasonable because the ratio of $R_{AB}$ to $R_S$ is nearly constant around its diurnal mean value
(Zhang et al., 2015b).
**(Fig. 2 here)**
*Synthesis of the carbon budget components*
Eddy covariance measured NEE is the difference between ecosystem carbon assimilation (i.e.,
GPP) and carbon release (i.e., ER), and the partitioning of NEE into GPP and ER constitutes
the first step of the carbon synthesis analysis. Ecosystem respiration originates from soil
heterotrophic respiration ($R_H$), below-ground autotrophic respiration ($R_{AB}$) (i.e., root
respiration) and above-ground autotrophic respiration ($R_{AA}$). By combining the eddy
covariance and soil respiration measurements, the carbon budget components can be derived
as follows.
The total soil respiration ($R_S$) is the sum of $R_H$ and $R_{AB}$,
$R_S=R_H+R_{AB}$.                                                                                  (6)
The total plant autotrophic respiration ($R_A$) is the difference between the ER from eddy
covariance measurement and $R_H$ from soil respiration measurement,



246 $R_A = ER - R_H.$       (7)

247 The above-ground autotrophic respiration ($R_{AA}$) is the difference between the ER from eddy

248 covariance measurement and $R_S$ from soil respiration measurement,

249 $R_{AA} = ER - R_S.$       (8)

250 NPP is the carbon stored in biomass, and can be calculated as the difference between GPP and

251 $R_A$,

252 $NPP_{EC} = GPP - R_A,$      (9)

253 where the subscript "EC" denotes that the NPP is based on the eddy covariance derived GPP.

254 In addition, NPP can also be inferred from crop samplings as the carbon storage in crops,

255 $NPP_{CS} = C_{cro},$       (10)

256 where the subscript "CS" denotes that NPP is based on crop samplings and carbon content

257 analysis, and the $C_{cro}$ is the carbon stored in crops.

258 NEP (also the inverse of NEE with eddy covariance) based on crop samplings is the

259 difference between the $NPP_{CS}$ and $R_H$ from soil respiration measurement,

260 $NEP_{CS} = NPP_{CS} - R_H.$      (11)

261 At this site, there are no disturbances of fire and insects, and no manure fertilization is

262 applied. The carbon input from seed is also negligible, and no crop straw is removed from the

263 field. Thus, the NBP can be estimated as the difference between NEP and carbon loss due to

264 grain export as,

265 $NBP = NEP - C_{gra},$      (12)



where $C_{gra}$ is grain carbon storage, NEP is estimated with two independent methods as
aforementioned, therefore, we also have two independent NBP estimates.



**Results**

*Environmental conditions, crop development and crop carbon content*

Fig. 3 show the seasonal variations in the environmental variables, including air temperature
with an average of 12.95 °C, vapor pressure deficit with an average of 0.70 kPa,
photosynthetic photon flux density with a yearly total of 7, 072.18 mol m$^{-2}$, precipitation with
a yearly total of 669.80 mm, groundwater table with an average of 2.15 m, soil moisture with
an average of 0.26 m$^3$ m$^{-3}$ and soil matric potential with an average of $-52.52$ kPa. The
seasonal maximum and minimum air temperature appear in July and January, respectively,
and vapor pressure deficit shows good accordance with air temperature. The groundwater
table fluctuation well follows irrigation event during winter and spring seasons, while follows
precipitation during summer and autumn seasons. In particular, the groundwater table ranges
from 0 to 3 m throughout the whole year. Water logging has happened in late August and early
September of the maize season, characterized by a very high groundwater table that is close to
0. The wet soil conditions prohibit this field from experiencing water stress (Fig. 3(d))
because even the lowest matric potential ($-187.6$ kPa) remains a lot higher than permanent
wilting point of crops (around $-1, 500.0$ MPa).
**(Fig. 3 here)**
Fig. 4 shows the seasonal evolution of canopy height and LAI as indicators of crop
development. The maximum LAI are 4.2 and 3.6 m$^2$ m$^{-2}$ for wheat and maize, respectively.
The variations in the $H_C$ and LAI well reflect the different stages of crop development. During
the wheat season, the start of the stages of regreening, jointing, booting, heading, and maturity





are approximately at March 1, April 20, May 1, May 7 and June 5, respectively. The different
crop stages agree well with the seasonal variations in biomass (Fig. 5), which shows that
wheat biomass accumulation mainly takes place in April and May, while maize biomass
accumulation mainly takes place in July and August. The total dry matter are 1, 717.5 g m$^{-2}$
for wheat and 1, 262.4 g m$^{-2}$ for maize at harvest, when wheat biomass are distributed as
follows: 3.0 % root, 42.7 % stem, 9.3 % leaf and 45.0 % grain; While maize biomass are
distributed as follows: 2.3 % root, 29.2 % stem, 7.1 % green leaf, 4.6 % dead leaf, 4.0 %
bracket, 7.3 % cob and 45.5 % grain. The averaged carbon contents of root, stem, green leaf,
dead leaf and grain are 410.4, 439.4, 486.0, 452.0 and 457.5 gC kg$^{-1}$ DM for wheat and,
407.7, 437.8, 477.2, 457.0 and 455.5 gC kg$^{-1}$ DM for maize (Table 1).
**(Table 1 here).**
**(Figs. 4 and 5 here)**
***Seasonal variations in the carbon budget components***
In this section, GPP is presented in negative values to indicate carbon removal from the
atmosphere. The seasonal variations in NEE, GPP and ER all follow bimodal curve patterns
corresponding with the two crop seasons (Fig. 6). All the three fluxes are almost in phase,
with peak appearing at the start of May during the wheat season and in the middle of August
during the maize season.
During the wheat dormant season, the carbon exchange is weak with the atmosphere, but the
cropland still absorbs carbon during most of the dormant season. The rotation periods
between two crops and the start of maize season are the main carbon source periods,



especially the start of maize season when the crop is tiny and the high temperature greatly
favors respiration. During the wheat season, two evident spikes appear on April 21st and May
8th with positive NEE (i.e., net carbon release), because the inclement weather suppressed
crop metabolism rate, similar phenomena also appear during the maize season. During the
rotation period between two crops, one evident spike also appears (characterized by a very
high ER of ~10 gC m$^{-2}$ d$^{-1}$) as a result of wheat residue decomposition following the rain
event.
**(Fig. 6 here)**
Fig. 7 shows the seasonal variation in ecosystem respiration and its components. During the
wheat season, the variation in ER follows crop development and temperature, but there are
two evident declines at the end of April and the start of May due to the low temperature
associated with inclement weather. During the early growing stage of maize, soil
heterotrophic respiration is the main component of ER. When water logging occurred in
August and September, both the soil heterotrophic respiration and below-ground autotrophic
respiration were suppressed to zero.
**(Fig. 7 here)**
***Seasonal total carbon budget***
Carbon flow shows that this wheat-maize rotation cropland has great potential to harvest
carbon from the atmosphere (Fig. 8). The seasonal total NEE, GPP and ER are −437.9, 1078.2
and 640.4 gC m$^{-2}$ for wheat, and −238.8, 779.7 and 540.8 gC m$^{-2}$ for maize. The NPP are
749.9 and 814.7 gC m$^{-2}$ for wheat based on crop sampling and the eddy covariance



complemented with soil respiration measurements, and are 591.6 and 531.9 gC m$^{-2}$ for maize
based on the two methods. Considering carbon loss in the form of soil heterotrophic
respiration, the NEP are 373.1 and 437.9 gC m$^{-2}$ for wheat based on the crop sampling and
eddy covariance measurement, and are 299.4 and 238.8 gC m$^{-2}$ for maize based on the two
methods. Furthermore, the carbon loss due to grain export are 346.7 and 265.2 gC m$^{-2}$ for
wheat and maize, respectively. Therefore, the NBP are 26.4 and 91.2 gC m$^{-2}$ for wheat based
on the two methods, and are 34.2 and −26.4 gC m$^{-2}$ for maize based on the two methods. We
finally take the average of two methods as estimates of NPP, NEP and NBP, which are 782.3,
405.5 and 58.8 gC m$^{-2}$ for the wheat season, and 561.8, 269.1 and 3.9 gC m$^{-2}$ for the maize
season. Considering the net carbon loss of −49.9 gC m$^{-2}$ during two rotation periods, NBP of
the whole wheat-maize crop cycle is 12.8 gC m$^{-2}$ yr$^{-1}$, indicating that the cropland is a weak
carbon sink.
**(Fig. 8 here)**
**Discussions**
*Comparisons with other croplands*
At the global scale, cropland is generally suggested as carbon neutral to the atmosphere (e.g.,
Ciais et al., 2010). However, numerous studies reported cropland as a carbon source (Anthoni
et al., 2004a; Verma et al., 2005; Kutsch et al., 2010; Wang et al., 2015; Eichelmann et al.,
2016), complemented with a few studies reporting cropland as a sink (e.g., Kutsch et al.,
2010). Such inconsistency probably results from different crop types, management intensities,
climatic conditions (Béziat et al., 2009; Smith et al., 2014) and fallow period length (Dold et



al., 2017). Our results demonstrate this fully irrigated wheat-maize rotation cropland featured
by a high groundwater table, is a weak carbon sink with NBP of 12.8 gC m$^{-2}$ under the field
condition. But other studies reported that this region is a carbon source (Li et al., 2006; Wang
et al., 2015). The difference probably originates from the different cropland managements and
soil conditions. In particular, our site experienced water logging because of the full irrigation
by water withdrawal from the Yellow River and the high precipitation of maize season. This
distinct field condition suppresses soil carbon loss in maize season (Fig. 7), potentially
converting the cropland from a carbon source to a sink. Because previous efforts show that
this site was a carbon source (Lei and Yang, 2010) without water logging happening in the
period from 2007 to 2008. The water logging event is occasionally reported in upland
croplands, Terazawa et al. (1992) and Iwasaki et al. (2010) found water logging cause damage
to plants, potentially explaining GPP decline in Dold et al. (2017) and also our study. While
our study further implies that water logging diminishes ecosystem respiration even more,
therefore reduces overall cropland carbon loss. However, more field control experiments and
modeling works remain required to further investigate how irrigation impacts cropland carbon
budget.
Comparing with another study at Luancheng site reporting North China Plain as a carbon
source (Wang et al., 2015), we found their estimates of GPP (1051 gC m$^{-2}$) and ER (692 gC
m$^{-2}$) in wheat season are pretty to our results (GPP of 1078.2 gC m$^{-2}$, and ER of 640.4 gC m$^{-}$
$^{2}$), such resemblance probably attributes to irrigations that prohibit both wheats from
experiencing water stress. However, maize of two studies exhibit considerable different





carbon fluxes. In particular, their GPP (984 gC m$^{-2}$) is a little higher than our result (779.7 gC
m$^{-2}$), but ER (841 gC m$^{-2}$) is a lot higher than our result (540.8 gC m$^{-2}$). The partitioning of
ER into three components, also exhibit contrasting features in these two studies, because
Wang et al. (2015) reported a relatively higher $R_{AA}$ (411 gC m$^{-2}$ for wheat and 428 gC m$^{-2}$ for
maize) that are more than three times of our study (128.0 gC m$^{-2}$ for wheat and 133.2 gC m$^{-2}$
for maize); But their relatively lower $R_{AB}$ (36 gC m$^{-2}$ for wheat and 16 gC m$^{-2}$ for maize) are
less than one quarter of our study (135.5 gC m$^{-2}$ for wheat and 115.4 gC m$^{-2}$ for maize); Their
$R_H$ of wheat (245 gC m$^{-2}$) is less than our estimate (376.8 gC m$^{-2}$), but $R_H$ of maize (397 gC m$^{-}$
$^{2}$) is greater than our result (292.2 gC m$^{-2}$). Such independent cross-site evaluations
demonstrate that carbon budget components may subject to specific cropland managements,
and even the same crop type can have diverse carbon behaviors under similar climatic
conditions. As aforementioned, the groundwater table is very high at our Weishan site because
the irrigation water is withdrawn from the Yellow River, but the Luancheng site in Wang et al.
(2015) is groundwater-fed with a very low groundwater table (around 42 m) (Shen et al.,
2013), featuring the major difference between these two sites. The water logging event and its
associated high soil moisture regimes at our site, contribute to both lower GPP and ER in
maize season. The lowered ER magnitude outweighs that of GPP, which eventually turns our
maize to a carbon sink. In contrast, Verma et al. (2005) reported that irrigation has turned a
maize cropland from carbon sink to source, because irrigation increased corn production that
is eventually exported from the cropland. However, no consensus has been reached on how
irrigation impacts cropland carbon behavior, but the emerging contrasting results point to the
necessity of investigating irrigation induced carbon budget change.





Our annual total NPP of 1, 344.1 gC m$^{-2}$ yr$^{-1}$ is almost double 714.0 gC m$^{-2}$ yr$^{-1}$ - the
approximate average of the model-estimated NPP for Chinese croplands with a rotation index
of 2 (i.e., two cropping cycles within one year) (Huang et al., 2007), and more than three
times the approximate 400 gC m$^{-2}$ yr$^{-1}$ estimated with MODIS (Zhao et al., 2005), and also a
little higher than 1, 144 gC m$^{-2}$ yr$^{-1}$ of a similar crop rotation at Luancheng site (Wang et al.,
2015). The higher NPP of this study site may partially result from the sufficient irrigation and
fertilization (Huang et al., 2007; Smith et al., 2014).
The carbon contents of wheat are comparable to the average value of 430 gC kg$^{-1}$ DM for
another wheat (Moureaux et al., 2008). The carbon content of different organs in maize show
different features from other maize cropland (e.g., Jans et al., 2010), which shows carbon
contents of the root, stem, leaf and corn were 316, 252, 452 and 468 gC kg$^{-1}$ DM (converted
from the unit of %), and the carbon contents of root and stem are clearly lower than our
results. These contrasting results suggest that the carbon content of the same crop may depend
on climate and environmental conditions. But the carbon contents of different organs for both
crops largely fluctuated across the season, implying the different temporal carbon features
associated with the crop stages.
***Carbon budget features in different ecosystems***
At the global scale, the carbon use efficiency (i.e., NPP/GPP) of crops (Table 2) is relatively
higher than both the average of 0.53 of forest (the slope of NPP against GPP, Delucia et al.
(2007)) (also see the examples of Griffis et al. (2004), Jassal et al. (2007) and Wu et al. (2013)
in Table 2) and, the average of 0.52 of terrestrial ecosystems (Zhang et al., 2009). In



particular, comparisons with the literature in Delucia et al. (2007) show that cropland is more
efficient in harvesting $CO_2$ from the atmosphere than forest. The carbon use efficiency of our
site (0.73 for wheat and 0.72 for maize) is higher than the average of 0.58 for croplands (Zhao
et al., 2005) and also higher than most other croplands of the same crops (e.g., 0.54 of a wheat
cropland in Moureaux et al. (2008), 0.45 and 0.56 of wheat in Aubinet et al. (2009), 0.55 of a
wheat in Suleau et al. (2011), 0.57 of a wheat and 0.55 of a maize in Wang et al. (2015), 0.51
and 0.35 of a wheat and maize in Demyan et al. (2016)). Considering the intense cropland
management at our site, these results imply that the intense management of sufficient
irrigation and fertilization may contribute to the higher carbon use efficiency. The carbon use
efficiencies of our study are comparable with the chickpea (0.74), sorghum (0.70), sunflower
(0.68) and wheat (0.77) in Albrizio and Steduto (2003), the consistent high carbon use
efficiency of various species of these two sites, indicate that carbon use efficiency levels are
regulated by both local site-specific microclimates and management types.
The different respiration partitionings of the same crop in different regions (e.g., wheat in our
study compared with Moureaux et al. (2008), Aubinet et al. (2009), Suleau et al. (2011), Wang
et al. (2015) and Demyan et al. (2016)) (See Table 2) indicate that carbon behavior may also
subject to environmental conditions and management practices. In particular, the ratio of
heterotrophic respiration to ecosystem respiration ($R_H$/ER) is greater in our research, probably
resulting from the full irrigation and high groundwater table prohibiting the soil from water
stress. These are different from other sites with similar crops (e.g., Moureaux et al., 2008;
Aubinet et al., 2009; Suleau et al., 2011; Wang et al., 2015; Demyan et al., 2016) that show

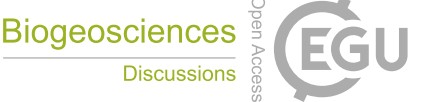

ecosystem respiration is dominated by below-ground and above autotrophic respirations. As
autotrophic respiration, especially above-ground autotrophic respiration in these studies
release high proportions of assimilated carbon by photosynthesis, therefore, their crops have
relatively lower carbon use efficiency as aforementioned.
**(Table 2 here)**
*Uncertainty in the estimation*
The NEE data from October 23[rd], 2010 to April 1[st], 2011, is calculated using SVR model
calibrated by previous measurements. The model performs well in predicting GPP and ER
with $R^2$ of 0.95 and 0.97, respectively. In addition, this data missing period was in the winter
time when carbon exchange accounts for a low percentage of the whole season, therefore this
estimate should have negligible effect on the annual carbon assessment.
The root biomass is difficult to measure, but the uncertainty should be limited as the root ratio
(the ratio of the root weight to the total biomass weight) accounts for only 10 % of the crop
(Jackson et al., 1996). The estimates of annual soil respiration are based on $Q_{10}$ model
validated by field measurements, which may bring about some uncertainty in soil respiration
budget because of the hysteresis response of soil respiration to temperature (e.g., Bahn et al.,
2008; Phillips et al., 2010; Zhang et al., 2015a). However, the $Q_{10}$ model remains robust in
soil respiration estimations if well validated (Tian et al., 1999; Latimer and Risk, 2016),
lending confidence to the estimates.
During the wheat season, the cumulative curve of $NPP_{EC}$ and $NPP_{CS}$ are not perfectly
consistent, because apparent differences have appeared during the winter wheat dormant



period from December 15$^{th}$, 2010 to March 8$^{th}$, 2011 (Fig. 9). These differences may result
from wheat sampling errors, as the sampling number is small compared to plant density.
However, the sampling at harvest is reliable because of the sufficient samples, and the two
NPPs at harvest showed no discernible difference, but the cumulative evolution of NPP$_{EC}$
agreed well with the NPP$_{CS}$ throughout the maize season (Fig. 9). Crop sampling and eddy
covariance methods provide consistently positive NBP estimates for wheat, implying that
wheat is a robust carbon sink. However, the two methods provide opposite results for maize,
with one showing maize is a carbon sink while the other showing a source. Though it remains
uncertain whether maize is a carbon sink or source, the average of these two results indeed
implies that maize is a weak sink. These results also indicate that field scale carbon budget
evaluation subjects to considerable uncertainties, again signifying this study and motivating
more efforts that improve carbon budget evaluation accuracy.
**(Fig. 9 here)**
**Conclusions**
The irrigated wheat-maize rotation cropland with high groundwater table in the North China
Plain, is a carbon sink with NBP of 12.8 gC m$^{-2}$ yr$^{-1}$. Most of the carbon sink happens in
wheat season with NBP of 58.8 gC m$^{-2}$, while maize is close to carbon neutral with NBP of
3.9 gC m$^{-2}$. The net carbon loss (49.9 gC m$^{-2}$) in the two rotation periods significantly
diminishes the carbon sink. The water logging appearing in maize season, contributes to
maintaining ecosystem carbon so that this cropland remains a carbon sink. The NPP are 782.3
and 561.8 gC m$^{-2}$ for wheat and maize, compensated by soil heterotrophic respiration, the



NEP are 405.5 and 269.1 gC m$^{-2}$ for wheat and maize. This cropland has high carbon use
efficiency (i.e., the NPP/GPP is 73 % and 72 % for wheat and maize, respectively), which
indicates this wheat-maize rotation cropland maintains a relatively higher proportion of
assimilated carbon via photosynthesis. The high $R_H$/ER (i.e., 59 % and 54 % for wheat and
maize, respectively) implies that soil heterotrophic respiration dominates ecosystem
respiration in this cropland. By comprehensively evaluating the carbon behavior of the
irrigated cropland with high groundwater table in the North China Plain, this study provides
valuable knowledge and perspectives of sustainable cropland management for mitigating
global carbon emission.





**Appendix A. Flux calculation of the period with equipment failure**

A1. Support Vector Regression method

Support Vector Regression (SVR) method is a machine-leaning technique-based regression, which transforms regression from nonlinear into linear by mapping the original low-dimensional input space to higher-dimensional space (Cristianini and Shave-Taylor, 2000). SVR method has two advantages: 1) the model training always converges to global optimal solution with only a few free parameters to adjust, and no experimentation is needed to determine the architecture of SVR; 2) SVR method is robust to small errors in the training data (Ueyama et al., 2013). The SVM software package obtained from LIBSVM (Chang and Lin, 2005) is used in this study.

A2. Data processing and selection of explanatory variables

Gross Primary Productivity (GPP) is influenced by several edaphic, atmospheric, and physiological variables, among which air temperature ($T_a$), relative humidity (RH), leaf area index (LAI), net photosynthetically active radiation (PAR), and soil moisture ($\theta$) are the dominant factors. Hence, we select $T_a$, RH, LAI, PAR, and $\theta$ as explanatory variables of GPP. Ecosystem Respiration (ER) consists of total soil respiration and above-ground autotrophic respiration, soil respiration is largely influenced by soil temperature and soil moisture, while above-ground autotrophic respiration is largely influenced by air temperature and above-ground biomass. So we select $T_a$, soil temperature at 5 cm ($T_{s5}$), $\theta$ and LAI as explanatory variables of ER. LAI is estimated from the Wide Dynamic Range Vegetation Index derived from the MOD09Q1 reflectance data (250 m, 8-d average,





https://lpdaac.usgs.gov/dataset_discovery/modis/modis_products_table/mod09q1, also see Lei
et al., 2013).
The three wheat seasons of 2005-2006, 2009-2010, and 2010-2011 are selected for model
training, and the original half-hourly measurements of GPP and ER together with the
explanatory variables are averaged to the daily scale, but we remove days missing more than
25 % of half-hourly data. We have a total of 466 GPP data samples and 483 ER samples for
model training. The explanatory variables for the equipment failure are also averaged into daily
scale, which will be used to calculate GPP and ER with the trained model described in the
following section.
A3. SVR model training and flux calculation
In order to eliminate the impact of variables with different absolute magnitudes, we rescale all
the variables in training-data set to the [0, 1] range prior to SVR model training. In the training
process, the radial basis function (RBF, a kernel function of SVR) is used and the width of
insensitive error band is set as 0.01. The SVR model training follows these steps:
(1) All training data samples are randomly divided into five non-overlapping subsets, and four
of them are selected as the training sets (also calibration set), the remaining subset is treated as
the test set (also validation set). Such process is repeated five times to ensure that every subset
has a chance to be the test set.
(2) For the selected training set, the SVR parameters (cost of errors c and kernel parameter σ)
are determined using a grid search with a five-fold cross-validation training process. In this
approach, the training set is further randomly divided into five non-overlapping subsets.



Training is performed on each of the four subsets within this training set, with the remaining
subset reserved for calculating the Root Mean Square Error (RMSE), and model parameters (c
and σ) yielding the minimum RMSE value are selected.
(3) The SVR model is trained based on the training set from step (1) and initialized by the
parameters (c and σ) derived from step (2).
(4) The test set from the step (1) is used to evaluate the model obtained from the step (3) by
using the coefficient of determination ($R^2$) and RMSE.
(5) The model is trained with all of the available samples, and the mean RMSE of GPP and ER
are 0.072 (gC m$^{-2}$ d$^{-1}$) and 0.048 (gC m$^{-2}$ d$^{-1}$), and $R^2$ are 0.95 and 0.97. GPP and ER are then
calculated with the trained model complemented with the observed explanatory variables
during equipment failure period, and NEE is derived as the difference of GPP and ER.
**Competing interest**
The authors declare that they have no conflict of interest.
**Acknowledgements**
This research was supported by the National Natural Science Foundation of China (Project
Nos. 51509187, 51679120 and 51525902), Tsinghua University Initiative Scientific Research
Program (2014z09112) and China Postdoctoral Science Foundation (No. 2015M570662).

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



**Table and figure**

Table 1 Carbon content of crop organs (gC kg$^{-1}$ DM)

| crop | date | root | stem | gree leaf | dead leaf | grain/corn |
|---|---|---|---|---|---|---|
| wheat | 3/15/2011 | 416 | 413 | 488 | - | - |
| | 3/22/2011 | 454 | - | 476 | - | - |
| | 3/29/2011 | - | 436 | 451 | - | - |
| | 4/5/2011 | 527 | 431 | 534 | - | - |
| | 4/13/2011 | 348 | 417 | 457 | - | - |
| | 4/21/2011 | 434 | 415 | 522 | - | - |
| | 4/29/2011 | 410 | 443 | 510 | - | - |
| | 5/6/2011 | 434 | 423 | 481 | - | - |
| | 5/14/2011 | 275 | 445 | 485 | - | - |
| | 5/22/2011 | 380 | 474 | - | 538 | 470 |
| | 5/29/2011 | 461 | 515 | 503 | 444 | 479 |
| | 6/5/2011 | 393 | 432 | 439 | 400 | 432 |
| | 6/10/2011 | 393 | 429 | - | 426 | 449 |
| maize | 7/4/2011 | 339 | 351 | 476 | - | - |
| | 7/13/2011 | 370 | 392 | 455 | - | - |
| | 7/21/2011 | 389 | 418 | 463 | - | - |
| | 7/29/2011 | 406 | 432 | 462 | - | - |
| | 8/5/2011 | 399 | 429 | 481 | - | - |
| | 8/12/2011 | 443 | 439 | 469 | - | - |
| | 8/22/2011 | 403 | 462 | 469 | - | - |
| | 9/3/2011 | 386 | 466 | 499 | - | 446 |
| | 9/11/2011 | 466 | 465 | 505 | - | 460 |
| | 9/20/2011 | 445 | 481 | 481 | - | 454 |
| | 9/30/2011 | 439 | 481 | 489 | 457 | 462 |





Table 2 Various ratios associated with carbon behaviors in different ecosystems

| plant type or species | NPP/GPP[a] | ER/GPP | $R_A$/GPP[a] | $C_{gra}$/NPP | $R_H$/ER | $R_{AB}$/ER | $R_{AA}$/ER | source |
|---|---|---|---|---|---|---|---|---|
| aspen | 0.54 | 0.76 | (0.46) | - | | (0.73)[b] | 0.27[c] | Griffis et al. (2004) |
| deciduous forest | 0.38 | 0.86 | 0.62 | - | 0.28 | 0.72[d] | | Wu et al. (2013) |
| douglas-fir | 0.47 | 0.86 | (0.53) | - | | (0.63)[b] | 0.37[c] | Jassal et al. (2007) |
| chickpea | 0.74 | - | (0.26) | - | - | - | - | Albrizio and Steduto (2003) |
| maize | 0.72 | 0.69 | 0.32 | 0.47 | 0.54 | 0.21 | 0.25 | **this study** |
| maize | 0.44 | 0.67 | 0.56 | - | 0.16 | 0.25 | 0.59 | Jans et al. (2010) |
| maize | 0.55 | 0.85 | 0.45 | 0.57 | 0.47 | 0.02 | 0.51 | Wang et al. (2015) |
| maize | (0.35) | 0.80 | 0.65 | - | 0.19 | 0.21 | 0.60 | Demyan et al. (2016)[e] |
| potato | 0.60 | 0.48 | 0.37 | 0.81[f] | 0.24 | 0.76 | | Aubinet et al. (2009)[g] |
| potato | (0.68) | 0.47 | 0.32 | - | 0.33 | 0.14 | 0.53 | Suleau et al. (2011) |
| sorghum | 0.70 | - | (0.30) | - | - | - | - | Albrizio and Steduto (2003) |
| sugar beet | 0.71 | 0.44 | 0.30 | 0.62[f] | 0.31 | 0.69 | | Aubinet et al. (2009)[g] |
| sugar beet | (0.78) | 0.36 | 0.22 | - | 0.37 | 0.25 | 0.36 | Suleau et al. (2011) |
| sunflower | 0.68 | - | (0.32) | - | - | - | - | Albrizio and Steduto (2003) |
| wheat | 0.73 | 0.59 | 0.24 | 0.44 | 0.59 | 0.21 | 0.20 | **this study** |
| wheat | 0.77 | - | (0.23) | - | - | - | - | Albrizio and Steduto, (2003) |
| wheat | 0.54 | 0.61 | 0.46 | - | 0.24 | 0.31 | 0.45 | Moureaux et al. (2008) |
| wheat (2005) | 0.56 | 0.60 | 0.44 | 0.42 | 0.26 | 0.74 | | Aubinet et al. (2009)[g] |
| wheat (2007) | 0.45 | 0.57 | 0.48 | 0.41 | 0.15 | 0.85 | | Aubinet et al. (2009)[g] |
| wheat | (0.55) | 0.57 | 0.45 | - | 0.21 | 0.17 | 0.62 | Suleau et al. (2011) |
| wheat | 0.57 | 0.66 | 0.43 | 0.45 | 0.35 | 0.05 | 0.59 | Wang et al. (2015) |
| wheat | (0.51) | 0.71 | 0.49 | - | 0.31 | 0.19 | 0.50 | Demyan et al. (2016)[e] |

Note: a- NPP+RA=GPP, we list both of NPP/GPP and $R_A$/GPP, the values in parentheses indicate that the value is calculated by the aforementioned closed equation.

Our study estimates NPP with two methods so that the equation is not closed, estimates in Aubinet et al. (2009) are not close either because they used different models to estimate respirations.

b- Tatio of total soil respiration to ecosystem respiration, i.e., $R_S$/ER or $(R_H + R_{AB})$/ER

c- Obtained as 1-$R_S$/ER

d- Ratio of autotrophic respiration to ecosystem respiration, i.e.,   $R_A$/ER=1-$R_H$/ER

e- The data is from 2012, and the estimation is based on the averaged carbon flux (ER and GPP) of both static and dynamic methods

f- The 'grain' production here is the sugar beet root production

g- Autotrophic respiraiton and heterotrophic respiraiton are averaged values of their two methods





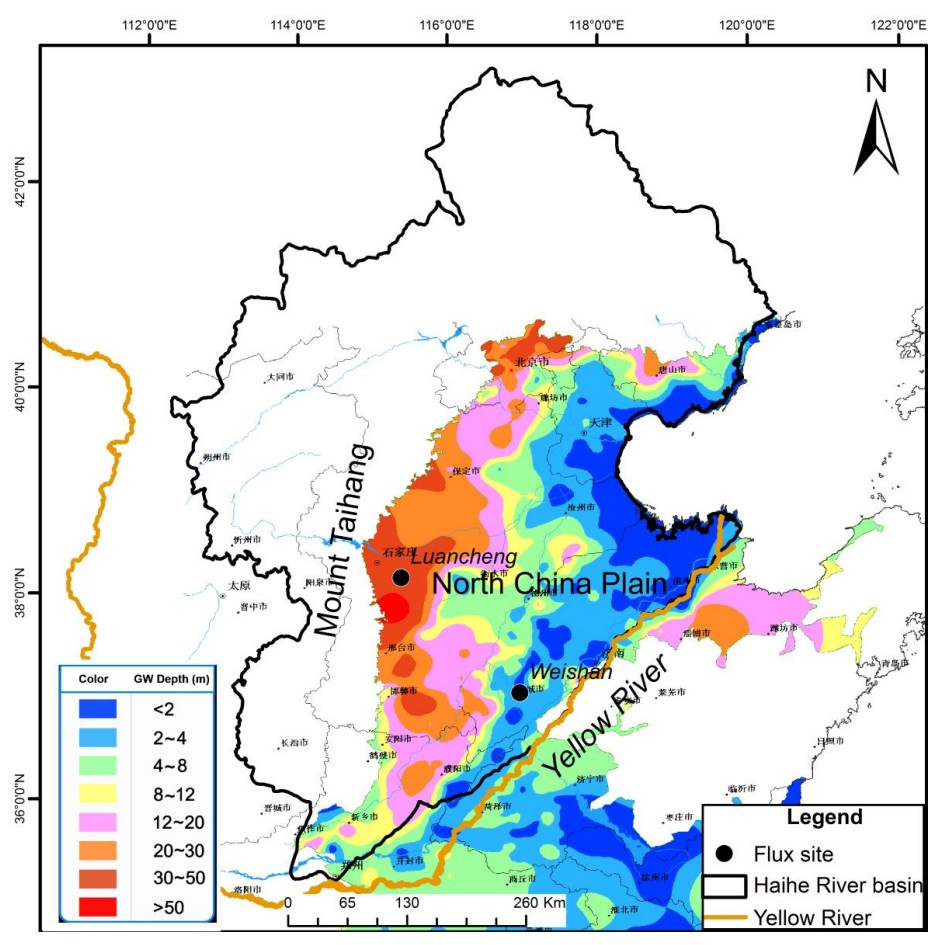


Fig. 1 Location of the experiment site. The background is the shallow groundwater depth in

early September of 2011 (source: http://dxs.hydroinfo.gov.cn/shuiziyuan/)






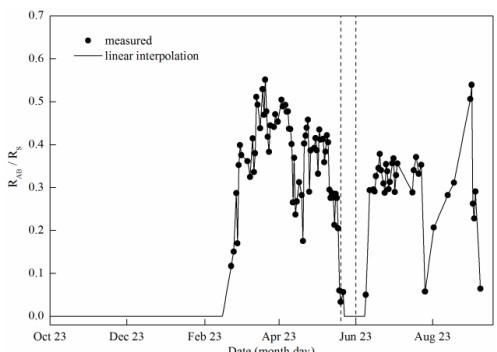


Fig. 2 Seasonal variations in the ratio of below-ground autotrophic respiration ($R_{AB}$) to total

soil respiration ($R_S$).







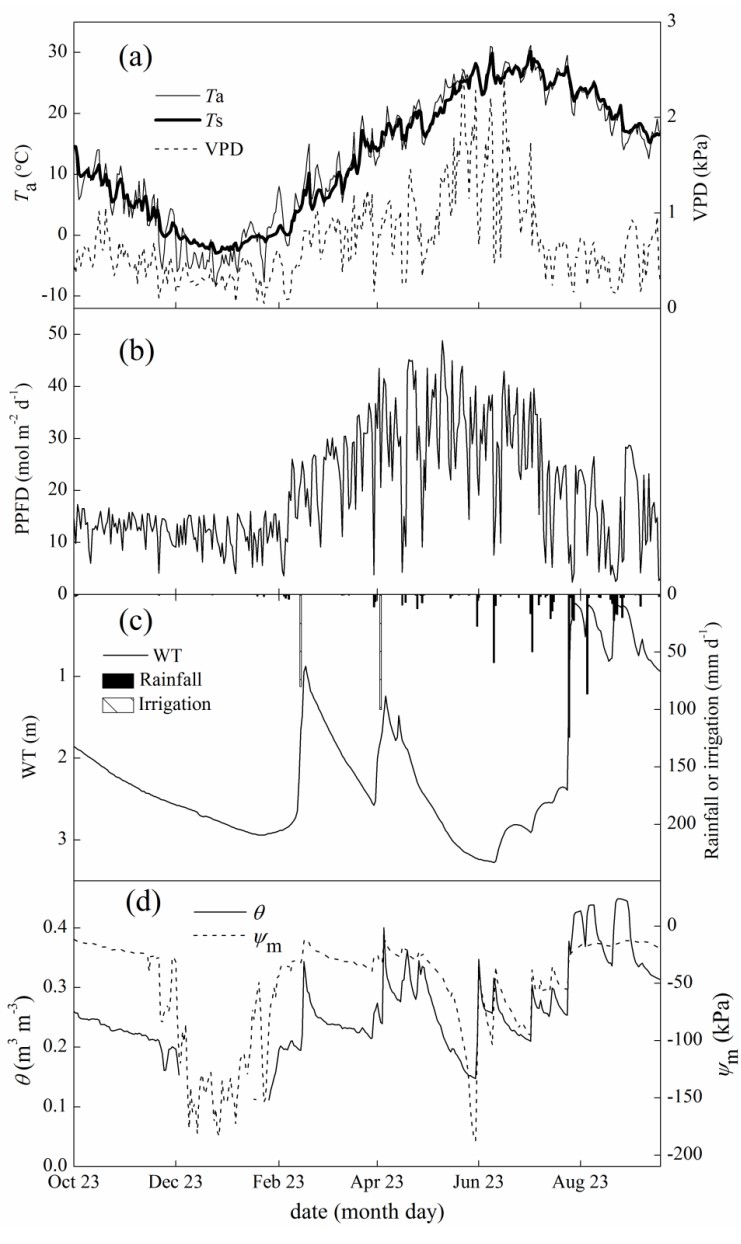


Fig. 3 Seasonal variations in the environmental variables of (a) air temperature ($T_a$) and vapor

pressure deficit (VPD), (b) photosynthetic photon flux density (PPFD), (c) rainfall, irrigation

and water table (WT) and (d) soil moisture ($\theta$) and soil matric potential ($\psi_m$).



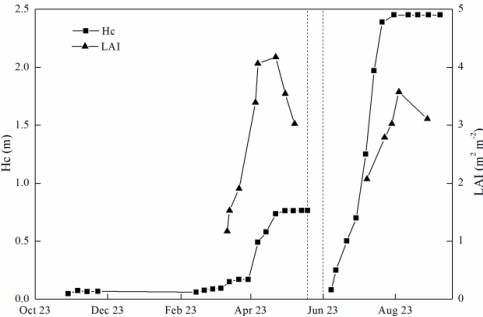


Fig. 4 Seasonal variations in canopy height ($H_C$), leaf area index (LAI). Two vertical dashed
lines (here and after) represent the date of harvesing wheat and sowing maize, respectively.

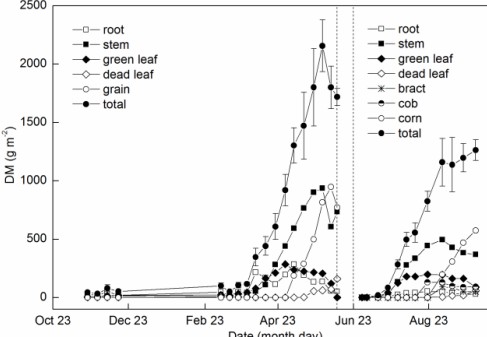


Fig. 5 Seasonal evolutions of dry biomass (DM). Different symbols denote different organs,
the error bar denotes 1 stardard deviation the four sampling points.

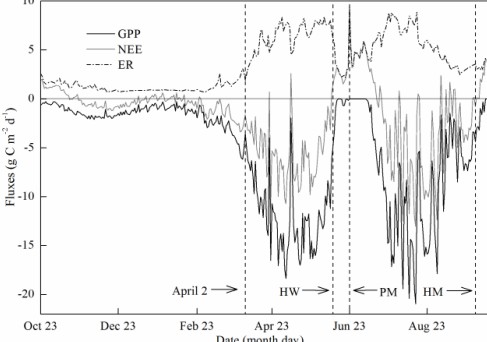


Fig. 6 Seasonal variations in gross primary producrivity (GPP), net ecosystem exchange
(NEE) and ecosystem respiration (ER) (Data before April 2 were calculated with SVR

method)



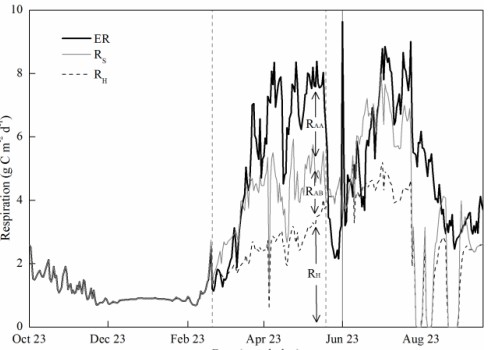


Fig. 7 Seasonal variations in the components of ecosystem respiration (ER), total soil
respiration ($R_S$), soil heterotrophic respiration ($R_H$). The difference between ER and $R_S$
denotes above-ground autotrophic respiration ($R_{AA}$), and the difference between $R_S$ and $R_H$

denotes below-ground autotrophic respiration ($R_{AB}$).


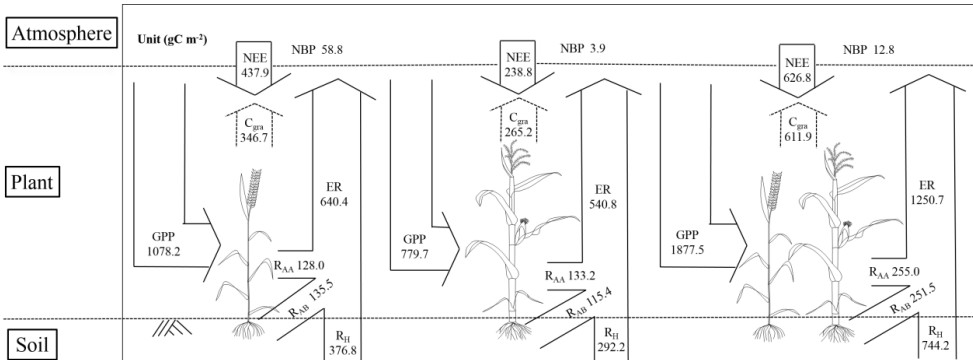


Fig. 8 Carbon budget of wheat (left), maize (middle) and the whole wheat-maize rotation
cycle (right) with rotation periods included. Note that NEE shown here is eddy covariance-

based measurements to maintain the carbon balance.





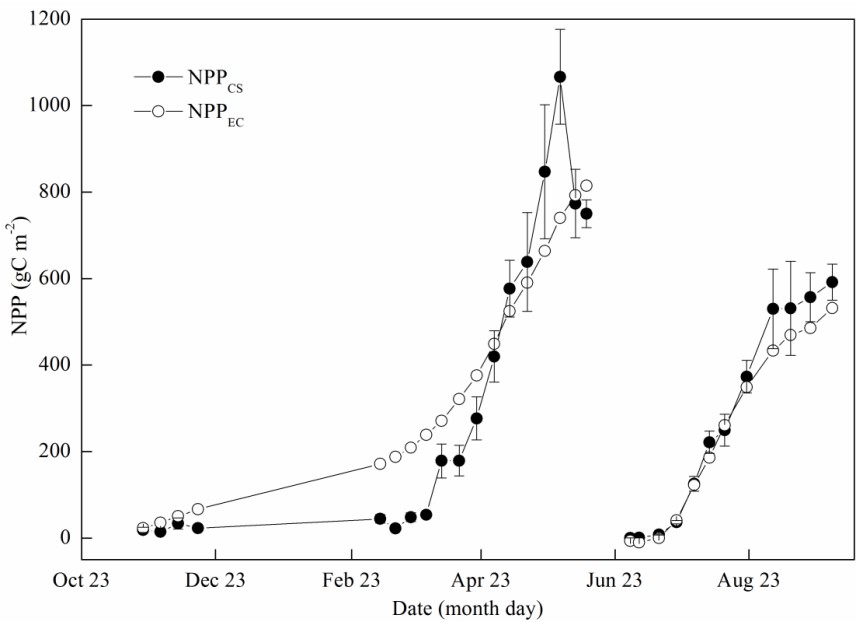


Fig. 9 Seasonal evolutions of the cumulative Net Primary Productivity (NPP) with two

independent methods of Crop Sampling (NPP$_{CS}$) and Eddy Covariance (NPP$_{EC}$)
complemented with soil respiration measurements.