# Peer review of "Carbon budget assessment of an irrigated wheat and maize rotation cropland with high groundwater table in the North China Plain"

_Biogeosciences, 2016_

## Referee Comment (RC1) · Anonymous Referee #1 · 5 Jan 2017

General comment : The authors present a detailed carbon budget of an irrigated wheat and maize rotation cropland that is characterized by high groundwater table due to irrigation systems. They report that the studied cropland behaves as a weak carbon sink. The paper is of general high quality with orginal results and sound data analyses, and the results are well discussed with the existing literature. However, I have a few concerns which I list here-below :

- The largest concern I have is with respect to the possibility that methane (CH4) emissions could be large in this type of poorly-drained and often flooded ecosystem. I think the authors should be a lot more cautious throughout their analysis, discussion and conclusion when they conclude that this ecosystem behaves as a carbon sink (even if

[Figure]

weak). This is only true with respect to CO2 but I am afraid that potentially large CH4 emissions could occur. Did you carry out CH4 emissions at this site ? Or could you provided any sort of estimation of these emissions ? Anyway, I strongly recommend to the authors to be more cautious with their conclusion about the carbon sink behaviour of this agro-ecosystem.

- The paper should be entirely proof-read for English spelling and type-setting mistakes. I provide here below the specific mistakes I could already notice.

- The authors paid attention to the evaluation of uncertainties associated to their measurements, which is a good point. However, I think that the uncertainties associated with the carbon budget terms (NBP, NEE, GPP, TER,...) are missing and should be estimated and given along with the mean values. They could indeed help to seize the reliability of claiming that the ecosystem behaves as a carbon sink.

- There are a few problems to fix in the figures (printed in black and white, some symbols or lines are not visible).

Specific comments

Abstract

L31-32 : Is this written as a general characteristic for croplands ? Or does this apply to this specific cropland ? This is not quite clear.

L32 : CUE should be defined (ratio of NPP to GPP).

Materials and methods

L182-183 : Is it really 200 wheat plants that are sampled at harvest ? This sounds really a lot.

L202 : can you check that reference temperature is well $0°$ ? Isn't it $10\ °C$, as it is often the case in the literature ?

Results

L338-342 : I think here the uncertainties related to the budget terms, and particularly NBP should be added, and particularly because the NBP values you give for wheat, maize and full crop rotation are averages.

Discussion

L357-359 (and in the conclusion) : I think here should appear some comments on the possibility that this ecosystem releases CH4 during those periods. Unless you can provide some measurements showing that no methane emissions are observed at this site ? At least more caution should be placed in this section.

L389-390 : I think this sentence is not correct : to me, GPP is the largest term and therefore it outweighs ER. Can you check this ?

L471 : As stated above, and unless it is certain that this site cannot be a source of CH4, I really think that some words should be added to say that this site could potentially be a net source of carbon if CH4 emissions are taken into account.

Tables and figures

Fig. 8 : As the NBP terms that appear on the figure are averages between two methods and do not correspond to the difference between NEE and ER, you should recall in the legend more clearly what is exactly this value of NBP.

Technical corrections

L68 : please remove the comma (,) after AND.

L96-98 : This sentence should be rewritten. Is one verb missing ?

L100 : please write leading TO dramatic. . .

L147-150 : This paragraph should be reformulated and the capital letters after the sign ; should be removed.

L178 : DISTRIBUTED rather than DISTRIBUTING

L196-200 : this sentence is too long and not very clear. This should be reformulated.

L270 : One S should be added to SHOW.

L283 : I think the correct term is -1500 kPa, and not -1500 MPa.

L294 : please remove the capital letter after the sign ;

L310 : please remove THE before HIGH TEMPERATURE

L363 : the word IN is missing before OUR STUDY

L363-365 : I do not understand this sentence, which should be re-written

L370 : I guess the term CLOSE should be added after PRETTY?

L375 : one S should be added to EXHIBIT

L378-379 : please remove the capital letter after the sign ;

L382 : BE should probably be added between MAY and SUBJECT

L431 : BE should be added between ALSO and SUBJECT

L437 : -GROUND should be added after ABOVE

L437-440 : I do not understand this sentence. Please can you reformulate ?

Table 1 : in the column titles, it should be written GREEN instead of GREE.

Fig. 3(c ): this figure is not clear, and particularly the distinction between rainfall and irrigation.

Fig.4 (legend) : one T is missing in the word HARVESING

Fig. 6 : The distinction between GPP and NEE colors and line types is not possible. In addition, the acronyms HW, PM and HM should be explained in the legend.

Fig. 5 (legend) : OF is missing before THE FOUR SAMPLING POINTS. And STAR-DARD should be written STANDARD. You can also add (SEE LEGEND IN GRAPH) after DIFFERENT ORGANS.
* * *

---

## Referee Comment (RC2) · Anonymous Referee #2 · 10 Jan 2017

A. General comments : This paper present a detailed carbon budget of a rotation wheat-maize rotation cropland with high groundwater table in North China. The idea of study a two cropping rotation by calculating all carbon budget components and the effect of high groundwater table on the carbon budget and its component is interesting and worthy of enquiry. They report that the NBP for the rotation is a weak carbon sink but the authors did not took into account the uncertainties associated to carbon budget terms. Also, methane emissions are not considered and could switch the plot from carbon sink to carbon source. Moreover, because the scientific questions are not explicitly announced, the authors didn't really comments the effect of groundwater tables on carbon budget, what is expected through the title of the paper. According to

the title, the reader expect a detailed analysis of the effect of groundwater levels on carbon budget and its components. Actually, some points need to be clarified in the material and method section as the flux treatment methodology but also some results (i.e. as about positives values of daily NEE on Fig. 5 ; see my comments below for the rest). Globally, a detailed description of flux treatment is missing (filtering, percentage of gapfilled data, . . .) but also the authors should use the equation described by Reichstein et al., 2005 to partition the flux instead of the one used in this paper. I think that a footprint analysis is missing particularly because two methods are compared and associated to calculate NBP. Also, biomass sampling is too small and non-representative of the plot. That leads to impossibility to compare the two methods (eddy covariance and sampling). Finally, ER terms are not spatially representatives of the whole plot (see below for details) and the too small sampling for biomass drove to some misinterpretations in the results and the following discussion & conclusion concerning ER and its components (RAB, RAA). For all these reasons, the study is fundamentally flawed in several different ways that make it unsuitable for publication.

B. Specific comments 1. Abstract (minors issues) ➢ L. 20 : modify the sentence as following ". . . from October 2010 to October 2011. During this cultural year, 2 crops were ground in the experimental period. For the winter wheat and summer-maize cycle, . . .". This is more explicit, according to me. ➢ Net biome productivity is calculated for a long-term period (C budget on several years) as the NEP (minimum one year, not a season). The integration periods are too short in this paper to talk about NBP and NEP when calculated for a season. ➢ L. 28 : what represents the term Âń Net ecosystem carbon release Âż ? ➢ L. 28 : There is only one rotation period, not two, but two crops in one cropping year. ➢ L. 29 : How was calculated the NBP term of the rotation cycle (12.8 gC m-2.yr-1) as NBP of wheat and maize were respectively 58.8 and 3.9 gC m-2.yr-1 ? ➢ L 31 : phrase pas supportée par les résultats de cet article. Idem dernière phrase : conclusion n'est pas supportée par les résultats de l'étude : pas de high CUE compare à d'autres cropland et pas forcément du à des effets ferti et irrigation. You cannot do CUE forte plus nfluencée par le fait qu'il y ait 2 cultures sur

une année plus que par les ferti et irr.

2. Introduction (minors issues) : 1) L. 50 : C budget components also includes C exported at harvest and C imported at sowing and through organic fertilisation (Ceschia et al., 2010). 2) Note : Use preferentially bibliographic reference in relation with cropland instead of forest (i.e. Ekblad et al., 2005) 3) L 65 : add Ceschia et al., 2010 in bibliography list (agro-ecosystem). 4) Freibauer et al., 2004 is missing in bibliography list. 5) L. 81 add Béziat et al., 2009 6) L. 84 add Verma et al., 2015 (suyker) 7) L 90 : add Ceschia et al., 2010 in bibliography 8) L. 92/93 : "More importantly, there remains no consensus on whether agro-ecosystem is a carbon sink or source" : Depending on management practices (i.e. choice of crops in the rotation, leaving crop residues, including cover crop in the rotation, irrigation practices, . . .), but also climatic year, agro-ecosystem will be C source or sink. There is no single answer for all agro-ecosystem. 9) L. 102 : is it correct to consider 5 m as a very high groundwater ?

3. Materials and methods (majors issues) : 1) L. 149 : The annual C budget have to be calculated on a full year, i.e from October 23rd 2010 to October 23rd 2011 (for 365 days). 2) L. 178 : 4 points/10 plant samples in wheat season and 3 plant samples in the maize season : globally, samples are too smalls. 3) LAI values are close to PAI values, is it really LAI? Or PAI (plant area index, meaning stem+leaf) ? or GLAI (green leaf area index) ? 4) Moreover, to compare the 2 methods to calculate NPP, it is necessary to collect the plant samples in the footprint or up from prevailing winds. Those informations are crucial and missing in this section. 5) L. 201 : The equation (1) is not the equation used by Reichstein et al., 2005 (neither Lei and Yang, 2010) as mentioned at L. 197. The equation used in your paper is not the best method, you should use the equation described by Reichstein et al., 2005. (Reichstein et al., 2005) Using this equation allow to better estimate the Rref that is temporally varying in an ecosystem. Rref has to be estimated for consecutive 4-day periods by nonlinear regression using the Lloyd & Taylor (1994) model, fixing all parameters except Rref. By this way, for each point in time, an estimate of Reco can be provided with time-dependent parameters and variables indicated by the symbol t in parentheses contrarily to the equation used in your paper (Reichstein et al., 2005). 6) It is essential that an uncertainty analysis is performed to quantify the uncertainty on the annual NEE (and maybe GPP and ER) values as it was done in Richardson & Hollinger (2007) for instance. Moreover, I would include a footprint analysis and an uncertainty analysis associated to footprint. An analysis using other cropping years would allow a direct comparison of the measurements and modelled data could give more credit to the analysis (see below). L. 206 "... the eddy covariance system failed from October 23rd 2010 to April 1st 2011": a comparison could be realised to estimate the uncertainty of the model compared to measured fluxes by eddy covariance and validate the use of the SVR model during this period (6 months) in your site. During the same period of the cropping year (before wheat sowing), the exercise aims at modelling fluxes with SVR model from October to April during bare soil period (for a cropping years with measured data) to compare them to measured fluxes. For example, the exercise is possible by modelling fluxes from October to April 2005-2006 by using 2009-2010 for model training. Then, those results would be compared with measured fluxes from October to April 2005-2006. By this way, you will be able to compare measured fluxes with modelled fluxes and estimate the uncertainties of the model to predict fluxes at this site and during this period. L. 515, "we have a total of 466 GPP data samples and 483 ER samples" : is that a total of half-hourly or daily data ? With the term "data samples" we understand half-hourly data that is not sufficient. 7) L. 216 : "Rs" term is not introduced – done in L. 242. 8) L. 219 : 3 replicated pairs of comparative treatments are not enough to reduce the uncertainty associated with spatial variability. 9) L. 244 : where were realised portable soil respiration system measurements ? Those measurements are they contained in the footprint? This is necessary to be comparable to eddy covariance measurements. If no footprint analysis were realised, specify the direction of prevailing wind. Moreover, provide information concerning the threshold for considering that the fluxes are representative of the plot and what is the percentage of the data filtered by the footprint analysis. 10) L. 258 : "NEP (also the inverse of NEE with eddy covariance)" :

this definition is false. Net Ecosystem Production (NEP) is annual NEE cumulated. In other word, NEE data are integrated over one year (365 days) to obtain annual NEP (Ceschia et al., 2010). 11) L. 266 : Replace "grain carbon storage" by "the amount of C exported at harvest". Globally, a detailed description of flux treatment is missing (filtering, percentage of gapfilled data, . . .) but also the authors should use the equation described by Reichstein et al., 2005 to partition the flux. I think that a footprint analyses is missing particularly because two methods are compared and associated to calculate NBP. Also, biomass sampling is too small and non-representative of the plot. That leads to impossibility to compare the two methods (eddy covariance and sampling).

4. Results/Discussion/Conclusion (majors issues) : 1) L. 184 : Âń The crop organs are separated Âż : How the authors seperated the roots ? L. 294/295 : 3.0 % and 2.3 % root for wheat and maize, respectively. Those values seem underestimated. Did the authors compared thoses results ? 2) L. 272 : Replace "a yearly total of 7, 072.18 mol m-2" by a mean value. 3) L. 276 : Replace "The groundwater table fluctuation well follows irrigation event during winter and spring seasons, while follows precipitation during summer and autumn seasons" by "The groundwater table fluctuations follows well irrigation events during winter and spring seasons, while they follows precipitations during summer and autumn seasons". 4) Figure 5 : Positives values of daily NEE are not caused by low temperatures as lower temperatures during those periods were never below 10°c. It might be a filtering problem. Many tests allow to control flux quality data as described in by Foken et al., 2004; Foken & Wichura, 1996; Göckede et al., 2004. Explain or detail the reason of positive daily value of NEE during the full crop development when temperatures are never under 10°C. The whole methodology of flux treatment has to be revised. ➢ L. 302 : "In this section, GPP is presented in negative values. . . from the atmosphere" but in L. 328, the GPP is positive : "the seasonal total NEE, GPP and ER are -437.9, 1078.2 and 640.4 gC m-2 for wheat, and -238.8, 779.7 and 540.8 gC m-2 for maize". GPP + ER = NEE ⇒ -1078.2 + 640.4 = -437.9 ➢ L. 308 : replace "cropping rotation" by "fallow rotation" and simplify the sentence (L. 308-311) "The fallow periods are the main carbon source periods, especially as well as

the period at the start of maize season when the crop is tiny and the high temperature greatly favors respiration Âż. ➢ L. 312 & 319/321 : " inclement weather suppressed crop metabolism rate" There is no visible and drastic climatic variation in Fig. 3. It could be a filtering problem. Realise a thorough assessment of the data. ➢ L. 322/324 : "When water logging occurred in August and September, both the soil heterotrophic respiration and below-ground autotrophic respiration were suppressed to zero". In this case, ER = 100% meaning that RAB is huge : approx. 5 gC m2/day in at the end of August and beginning of September while before RAB represented only 1 gC m2/day. This example reflects the lack of spatial representativity of the measures between the two methods : ➢ Either there is a measurement problem for RH and RS when water logging occurred ➢ Or RH and RS are highly overestimated most of the time (by this way, those results would be in line with the results reported by Wang et al., 2015 and Moureaux et al., 2008 but also more consistent with aboveground/belowground biomass measurements cited before. This example also show that the footprint analysis is missing in this study because the measurements of ER and all its component has to be done in the footprint zone to be comparable to eddy covariance measurements. ➢ If you consider that negative values indicate carbon removal from the atmosphere, the NPP should also be negative. In L. 330, NPP values are positives. ➢ L. 333 : the C exported at harvest should be a positive term to indicate C back to the atmosphere as in L. 302 they said that the "GPP is a negative term to indicate C removal from the atmosphere". ➢ Whatever the definition of the sign convention chosen, ensure that this convention is respected in the whole document. ➢ Moreover, describe explicitly the sink/source effect of each component of the C budget as did Whang et al. (2015); cited in the paper (i.e. L. 65). ➢ Also add a table with chosen sign convention for each term as did in the Table 2 in Whang et al. (2015). ➢ L. 340 : How is calculated the "net carbon loss" ? ➢ L. 340-342 : There is only one rotation period, not two, but two crops in one cropping year. ➢ L. 350 : detail "management intensities" : straws removal or not, use of organic manure or not,… ➢ L 355 to 357 : Further develop ⇒ other differences : ➢ Straw management, organic manure? ➢ Strong differ-

ences for NEP terms between those several years associated to contrasted climatic years ? ➢ Differences in exportations (at harvest) terms ? ➢ L. 359 to 361 : I recommend further analysis of theses differences. ➢ L. 361 to 365 : "The water logging event is occasionally reported in upland croplands, Terazawa et al. (1992) and Iwasaki et al. (2010) found water logging cause damage to plants, potentially explaining GPP decline in Dold et al. (2017) and also our study. While our study further implies that water logging diminishes ecosystem respiration even more, therefore reduces overall cropland carbon loss. " ⇒ so it affects certainly the yield and the carbon budget ! Not discussed. ➢ L. 368 to 372 : "Comparing with another study at Luancheng site reporting North China Plain as a carbon source (Wang et al., 2015), we found their estimates of GPP (1051 gC m-2) and ER (692 gC m-2) in wheat season are pretty close to our results (GPP of 1078.2 gC m-2, and ER of 640.4 gC m 2), such resemblance probably attributes to irrigations that prohibit both wheats from experiencing water stress." ⇒ potentially many other causes : way too much focus on the effect of irrigation. ➢ L. 372 to 373 : "However, maize of two studies exhibit considerable different carbon fluxes. " ⇒ compare the components of the C budget with other studies for cropland (i.e. Moureaux et al., 2008). ➢ L. 387 : "…featuring the major difference between these two sites " ⇒ what are the others terms of management? ➢ L. 405 : replace "corn" by "grain" ➢ L. 412 : "NPP/GPP" not reliable in this study, root biomass estimates are probably very underestimated + too few samples. ➢ L. 432 : add "be" before "subject to environnemental…" ➢ L. 437 : "ecosystem respiration is dominated by below-ground and above autotrophic respirations" ⇒ this difference with the literature reveals a methodological problem in this study (see before for details). ➢ L. 449 : Âń 10% Âż for the root ratio compared to only 2.3 and 3% in this study. ➢ L. 460 : "because of the sufficient samples" ⇒ too low according to me. ➢ L. 469 : not enough samples for estimating NPP ! ➢ L. 473 : Calculate uncertainties for each component of NBP. Also NBP = 12.8 gC m-2 yr-1 is not significant because of uncertainties associated to the term. ➢ L. 475 : replace "rotation period" by "fallow period". ➢ L. 479 to 484 : "NEP are 405.5 and 269.1 gC m-2 for wheat and

maize. This cropland has high carbon use efficiency (i.e., the NPP/GPP is 73 % and 72 % for wheat and maize, respectively), which indicates this wheat-maize rotation cropland maintains a relatively higher proportion of assimilated carbon via photosynthesis. The high RH/ER (i.e., 59 % and 54 % for wheat and maize, respectively) implies that soil heterotrophic respiration dominates ecosystem respiration in this cropland. " ⇒I have some doubts concerning those conclusions given the small number of soil respiration measurements. ➢ In that state, it is difficult to evaluate the results of this study as signs conventions are not explicitly described. Moreover, the sign convention described in L.302 is opposite to the rest of the document. Whatever the sign convention adopted, please respected it in the whole document. Indeed, it is hard to understand which terms are C sources or C sinks and conclude about the results.

C. Technical corrections (non exhaustive) • Replace the ";" by "." in lines : 294, 378 • Add dots in units, i.e. replace "m m-3" by "m.m-3" (L. 274, L. 286) . . . • Fig. 3c : irrigation events are difficult to distinguish on the figure. • Figure 4 : add standard deviations. • Figure 6 : explain abbreviations HW. And PM.

---

## Author Comment (AC1) · 27 Feb 2017

Response to reviewer's comments on "Carbon budget assessment of an irrigated wheat and maize rotation cropland with high groundwater table in the North China Plain" by Quan Zhang et al.

We appreciate the reviewer for the constructive comments, which will take seriously and use to improve the manuscript.

Anonymous Referee #1, comments received and published 05 Jan 2017

General comment:

The authors present a detailed carbon budget of an irrigated wheat and maize rotation cropland that is characterized by high groundwater table due to irrigation systems. They report that the studied cropland behaves as a weak carbon sink. The paper is of general high quality with original results and sound data analyses, and the results are well discussed with the existing literature.

Response:

We appreciate the reviewer's support for the quality of the manuscript and the constructive comments.

However, I have a few concerns which I list here-below:

- The largest concern I have is with respect to the possibility that methane (CH4) emissions could be large in this type of poorly-drained and often flooded ecosystem. I think the authors should be a lot more cautious throughout their analysis, discussion and conclusion when they conclude that this ecosystem behaves as a carbon sink (even if weak). This is only true with respect to CO2 but I am afraid that potentially large CH4 emissions could occur. Did you carry out CH4 emissions at this site? Or could you provide any sort of estimation of these emissions? Anyway, I strongly recommend to the authors to be more cautious with their conclusion about the carbon sink behaviour of this agro-ecosystem.

Response:

We totally agree with the reviewer on that methane may be significant during the water logging condition. Because this is an upland cropland, methane issue has not been considered in previous efforts, therefore its measurement is not available at this experiment site. The reviewer is right that we should be cautious with the conclusion of 'carbon sink'. Considering $CO_2$ budget of this study remains a complete analysis, therefore we will change our topic from general carbon budget to $CO_2$ budget, which is more concise to this manuscript. We will change the title to "The budget of $CO_2$ in an irrigated wheat and maize rotation cropland over the North China Plain"

Following the reviewer's advice, we also realized methane release may be significant when soil is saturated, so we will definitely add methane measurement in future experiment for a complete carbon budget estimate.

- The paper should be entirely proof-read for English spelling and type-setting mistakes. I provide here below the specific mistakes I could already notice.

 Response:

We will revise the whole manuscript carefully and ask an English native for further edits.

- The authors paid attention to the evaluation of uncertainties associated to their measurements, which is a good point. However, I think that the uncertainties associated with the carbon budget terms (NBP, NEE, GPP, TER,: : :) are missing and should be estimated and given along with the mean values. They could indeed help to seize the reliability of claiming that the ecosystem behaves as a carbon sink.

Response:

We will add uncertainty analysis for eddy covariance based NEE, GPP and TER. We will use the 'successive days approach' proposed by Hollinger and Richardson (2005) to quantify the uncertainty of eddy covariance based carbon flux.

- There are a few problems to fix in the figures (printed in black and white, some symbols or lines are not visible).

Response:

We appreciate the reviewer point out this problem, and we will redraw these figures to present them more clearly.

Specific comments

Abstract

L31-32: Is this written as a general characteristic for croplands? Or does this apply to this specific cropland? This is not quite clear.

Response:

This characteristic is very common from the literature we have collected, see Table 2 in the manuscript. We realized this conclusion may be not strong, because the carbon use efficiency is also subject to managements of irrigation, fertilization etc. Hence, we will only compare the carbon use efficiency in the discussion section, whereas no present in the abstract.

L32: CUE should be defined (ratio of NPP to GPP).

Response:

Will revise.

Materials and methods

L182-183: Is it really 200 wheat plants that are sampled at harvest? This sounds really a lot.

Response:

Correct, we collected 200 wheat plants at harvest to reduce the uncertainty associated with plant samplings. Such big samplings allow us the confidence in NPP estimate.

L202: can you check that reference temperature is well 0 °C? Isn't it 10 °C, as it is often the case in the literature?

Response:

We got the reviewer's point.

The reference temperature depends on the selected temperature response function, $ER_{ref}$ here is the reference respiration at 0 °C because we use the function $ER = ER_{ref} \exp(bT_S)$.

But it should be reference respiration at 10 °C if the function writes as $ER = ER_{ref} Q_{10}^{(T_S - 10)/10}$. These two functions are equivalent *per se*, and are linked by $Q_{10}=\exp(10b)$. And the reference respiration should also be 10 °C when using the temperature response curve by Reichstein et al. (2005), i.e., $ER = ER_{ref} \exp[E_0(1/(T_{ref} - T_0) - 1/(T - T_0))]$.

Results

L338-342: I think here the uncertainties related to the budget terms, and particularly NBP should be added, and particularly because the NBP values you give for wheat, maize and full crop rotation are averages.

Response:

We will add uncertainty for the $CO_2$ budget components, and also evaluate the uncertainty of NBP estimate.

Discussion

L357-359 (and in the conclusion): I think here should appear some comments on the possibility that this ecosystem releases CH4 during those periods. Unless you can provide some measurements showing that no methane emissions are observed at this site? At least more caution should be placed in this section.

Response:

We realized that CH4 flux may be significant at our site during the water logging period, but we do not have such measurement, so we will focus our topic on $CO_2$ budget. A thorough estimate of $CO_2$ budget remains a necessary work for carbon cycle in cropland of this kind, but we will incorporate CH4 in future work.

L389-390: I think this sentence is not correct: to me, GPP is the largest term and therefore it outweighs ER. Can you check this?

Response:

The reviewer is right, GPP is usually the largest flux term.

Here we wanted to express that both GPP and ER decrease in water logging conditions, but the reduction of ER (i.e., ΔER=ER(no water logging)-ER(water logging)) is higher than that of GPP (ΔGPP=GPP(no water logging)-GPP(water logging)), so the net ecosystem carbon sink (i.e., GPP-ER) is higher in water logging conditions than under regular field conditions, if explaining it from a mathematical perspective,

we have ΔER> ΔGPP, i.e.,

ER(no water logging)-ER(water logging)> GPP(no water logging)-GPP(water logging)

then we get,

GPP(water logging)-ER(water logging) > GPP(no water logging)-ER(no water logging),

which indicates that net carbon sink is higher in water logging conditions.

We will revise this part to explain it more clearly.

L471: As stated above, and unless it is certain that this site cannot be a source of CH4, I really think that some words should be added to say that this site could potentially be a net source of carbon if CH4 emissions are taken into account.

Response:

We agree with the reviewer, because of lacking $CH_4$ measurement, we will focus this study on $CO_2$ and will modify our conclusion on $CO_2$ alone. However, we will incorporate the reviewer's comments by discussing a little about the possible emission of $CH_4$ as an outlook for future study.

Tables and figures

Fig. 8: As the NBP terms that appear on the figure are averages between two methods and do not correspond to the difference between NEE and ER, you should recall in the legend more clearly what is exactly this value of NBP.

Response:

We appreciate the review's advice, we will make it clearer by adding texts explicitly explaining that NBP is the average of two independent methods (i.e., eddy covariance-based and crop sampling-based).

Technical corrections

L68 : please remove the comma (,) after AND.

Response:

Will revise.

L96-98 : This sentence should be rewritten. Is one verb missing ?

Response:

The sentence was not clear, we will revise it into "The North China Plain is one of the most important food production regions in China, it guarantees the national food security by providing more than 50 % wheat and 33 % maize to the whole nation (Kendy et al., 2003).

L100 : please write leading TO dramatic: : :

Response:

Will revise.

L147-150 : This paragraph should be reformulated and the capital letters after the sign ; should be removed.

Response:

Will reformulate this paragraph, and use lower case word after sign ';'.

L178 : DISTRIBUTED rather than DISTRIBUTING

Response:

Will revise.

L196-200 : this sentence is too long and not very clear. This should be reformulated.

Response:

We will revise this part to describe the details of NEE partitioning method, we used the method proposed by Reichstein et al. (2005) and we will describe the details.

L270 : One S should be added to SHOW.

Response:

Will revise.

L283 : I think the correct term is -1500 kPa, and not -1500 MPa.

Response:

The reviewer is right, the unit is messed up. We will revise this.

L294 : please remove the capital letter after the sign ;

Response:

Will revise.

L310 : please remove THE before HIGH TEMPERATURE

Response:

Will revise.

L363 : the word IN is missing before OUR STUDY

Response:

Will revise.

L363-365 : I do not understand this sentence, which should be re-written

Response:

We wanted to express that ecosystem respiration decline magnitude is higher in water logging condition than the regular field condition, so the carbon loss from filed was suppressed in water logging conditions.

We will revise the sentence into some explanation like "While our study further implies that water logging condition suppresses Ecosystem Respiration more than Gross Primary Productivity, therefore reduces the net $CO_2$ release to the atmosphere"

L370 : I guess the term CLOSE should be added after PRETTY?

Response:

Correct, will revise.

L375 : one S should be added to EXHIBIT

Response:

Will revise.

L378-379 : please remove the capital letter after the sign ;

Response:

Will revise.

L382 : BE should probably be added between MAY and SUBJECT

Response:

Will revise.

L431 : BE should be added between ALSO and SUBJECT

Response:

Will revise.

L437 : -GROUND should be added after ABOVE

Response:

Will revise.

L437-440 : I do not understand this sentence. Please can you reformulate ?

'ecosystem respiration is dominated by below- ground and above autotrophic respirations. As autotrophic respiration, especially above-ground autotrophic respiration in these studies release high proportions of assimilated carbon by photosynthesis, therefore, their crops have relatively lower carbon use efficiency as aforementioned.'

Response:

We will revise this to explain more clearly.

We wanted to explain the reason why previous studies report a lower carbon use efficiency (i.e., CUE=NPP/GPP). Because NPP=GPP-RA=GPP-(RAB+RAA), then we get CUE=1-(RAB+RAA)/GPP. The previous studies have a high (RAB+RAA)/GPP ratio as an index of carbon release through autotrophic respiration, supporting the reason why they have lower carbon use efficiency. Here RA-autotrophic respiration; RAB- below-ground autotrophic respiration; RAA- above-ground autotrophic respiration.

Table 1: in the column titles, it should be written GREEN instead of GREE.

Response:

Will revise.

Fig. 3(c): this figure is not clear, and particularly the distinction between rainfall and irrigation.

Response:

We will redraw this figure to make it clearer.

Fig.4 (legend): one T is missing in the word HARVESING

Response:

Will revise.

Fig. 6: The distinction between GPP and NEE colors and line types is not possible. In addition, the acronyms HW, PM and HM should be explained in the legend.

Response:

We will redraw this figure to make it clearer.

We will also add explanation of HW, PM and HM in the figure caption. They are: HW-Harvest Wheat, PM-Plant Maize, HM-Harvest Maize

Fig. 5 (legend): OF is missing before THE FOUR SAMPLING POINTS. And STARDARD should be written STANDARD. You can also add (SEE LEGEND IN GRAPH) after DIFFERENT ORGANS.

Response:

Will revise.

References used in this response

Kendy, E., Gerard-Marchant, P., Walter, M. T., Zhang, Y. Q., Liu, C. M., and Steenhuis, T. S.: A soil-water-balance approach to quantify groundwater recharge from irrigated cropland in the North China Plain, Hydrol. Process., 17, 2011-2031, doi: 10.1002/hyp.1240, 2003.

Reichstein, M., Falge, E., Baldocchi, D., Papale, D., Aubinet, M., Berbigier, P., Bernhofer, C., Buchmann, N., Gilmanov, T., Granier, A., Grunwald, T., Havrankova, K., Ilvesniemi, H., Janous, D., Knohl, A., Laurila, T., Lohila, A., Loustau, D., Matteucci, G., Meyers, T., Miglietta, F., Ourcival, J. M., Pumpanen, J., Rambal, S., Rotenberg, E., Sanz, M., Tenhunen, J., Seufert, G., Vaccari, F., Vesala, T., Yakir, D., and Valentini, R.: On the separation of net ecosystem exchange into assimilation and ecosystem respiration: review and improved algorithm, Global Change Biol., 11, 1424-1439, doi: 10.1111/j.1365-2486.2005.001002.x, 2005.

---

## Author Comment (AC2) · 27 Feb 2017

Response to reviewer's comments on "Carbon budget assessment of an irrigated wheat and maize rotation cropland with high groundwater table in the North China Plain" by Quan Zhang et al.

We appreciate the reviewer for the constructive comments, which we will take seriously and use to improve the manuscript.

Anonymous Referee #2, comments received and published 10 Jan 2017

A. General comments

This paper present a detailed carbon budget of a rotation wheat-maize rotation cropland with high groundwater table in North China. The idea of study a two cropping rotation by calculating all carbon budget components and the effect of high groundwater table on the carbon budget and its component is interesting and worthy of enquiry.

Response:

We appreciate the reviewer's positive comments to our work. We also appreciate the reviewer's efforts to help improve the manuscript.

They report that the NBP for the rotation is a weak carbon sink but the authors did not took into account the uncertainties associated to carbon budget terms. Also, methane emissions are not considered and could switch the plot from carbon sink to carbon source. Moreover, because the scientific questions are not explicitly announced, the authors didn't really comments the effect of groundwater tables on carbon budget, what is expected through the title of the paper. According to the title, the reader expect a detailed analysis of the effect of groundwater levels on carbon budget and its components.

Response:

We will add uncertainty analysis for eddy covariance measured NEE, GPP and ER using the 'successive days approach' proposed by Hollinger and Richardson (2005). Reviewer#1 also mentioned the methane issue, but we will focus on the $CO_2$ budget due to lacking methane measurement, please also see the response to reviewer#1.

The high groundwater table is a special feature of this upland cropland, so we wanted to convey this information to the readers in title. However, our goal is to provide a complete estimate of $CO_2$ budget of this typical irrigated cropland. We realized our title may lead the readers to a fault direction, so we will change the title into "The budget of $CO_2$ in an irrigated wheat and maize rotation cropland over the North China Plain", we will also revise the introduction section to clearly announce the topic.

Actually, some points need to be clarified in the material and method section as the flux treatment methodology but also some results (i.e. as about positives values of daily NEE on Fig. 5; see my comments below for the rest).

Response:

We will follow the reviewer's advice to describe the details of flux treatment, and the result of positive NEE has been addressed in the response to the specific comments.

Globally, a detailed description of flux treatment is missing (filtering, percentage of gapfilled data) but also the authors should use the equation described by Reichstein et al., 2005 to partition the flux instead of the one used in this paper.

Response:

We did not present too much of the flux treatment, because the procedure can be found elsewhere (Lei and Yang, 2010a), so we would like to focus on the overall experiment design. However, by following the reviewer's advice, we will add more description on flux data treatment, which includes flux filtering, QA/QC, gap filling. For NEE partitioning, we actually used the method proposed by Reichstein et al. (2005), but we have not presented them clearly in order to reduce the text, we will describe the details.

I think that a footprint analysis is missing particularly because two methods are compared and associated to calculate NBP. Also, biomass sampling is too small and non-representative of the plot. That leads to impossibility to compare the two methods (eddy covariance and sampling).

Response:

Footprint analysis has been done for this site before (Lei and Yang, 2010b), it shows "At the unstable condition, our analysis indicated that approximately 90% of the measured flux was expected to come from within the nearest 420 and 166 m during wheat and maize periods, respectively", and we took that as a guide to carry out all the experiments that fell in the footprint. More importantly, our field is very flat and the crop canopy is very homogeneous (see Fig. 1 as an example of wheat season), so we think it unnecessary to redo the footprint analysis, but we will definitely assure the readers that all our experiments were conducted in the footprint by adding such description.

[Figure]

Fig. 1 The field condition right before wheat harvest at the experiment site

Finally, ER terms are not spatially representatives of the whole plot (see below for details) and the too small sampling for biomass drove to some misinterpretations in the results and the following discussion & conclusion concerning ER and its components (RAB, RAA). For all these reasons, the study is fundamentally flawed in several different ways that make it unsuitable for publication.

Response:

The experimental field is very flat and homogeneous, the eddy covariance and soil respiration measurements are representative. It is a common method to combine eddy covariance and soil respiration measurements within footprint to partition ER (e.g., Moureaux et al., 2008), in the current stage, it remains a practical way to partition ER. Our field and crop are very homogeneous, which allow us the confidence of representativeness.

As to the biomass sampling, it was indeed small during the main growing season, but sampling of this period was not used for NPP estimate, therefore the small sampling during growing season has nothing to do with NPP estimate. However, we sampled 200 plants for wheat and 5 plants for maize at each of the 4 plots at harvest, when the biomass sampling is used for NPP estimate. The samplings are representative because the crops are very homogeneous, reflected by the acceptable standard deviation of four sampling plots (Fig. 5 in the manuscript).

For these reasons described above, we think we will solve all the questions from the reviewer's perspectives. This work has contributed to $CO_2$ budget research and deserves publishing, because it will allow a direct estimate of $CO_2$ budget for this typical irrigated cropland in North China Plain.

B. Specific comments

1. Abstract (minors issues)

L. 20: modify the sentence as following ": : : from October 2010 to October 2011. During this cultural year, crops were ground in the experimental period. For the winter wheat and summer-maize cycle, : : :". This is more explicit, according to me. Net biome productivity is calculated for a long-term period (C budget on several years) as the NEP (minimum one year, not a season). The integration periods are too short in this paper to talk about NBP and NEP when calculated for a season.

Response:

We agree with the reviewer on that long-term evaluation of NBP and NEP will provide more valuable estimate. However, our goal is to investigate the $CO_2$ budget and all of its

components, i.e., GPP, ER, RH, RAB and RAA. We reported our estimate of NEP and NBP, because the detailed $CO_2$ components allow for such estimations, which definitely will give the readers a direct knowledge of the $CO_2$ characteristics at this typical cropland. For these, we think it remains important to report the NBP and NEP, though we just have one year's estimate.

L. 28: what represents the term Net ecosystem carbon release?

Response:

It means the cropland $CO_2$ release to the atmosphere during fallow period, because there is no plant in the field and NEE are always positive at this period. We will revise this part to 'total $CO_2$ loss' to make it clearer.

L. 28 : There is only one rotation period, not two, but two crops in one cropping year.

Response:

Will revise.

L. 29 : How was calculated the NBP term of the rotation cycle (12.8 gC m-2.yr-1) as NBP of wheat and maize were respectively 58.8 and 3.9 gC m-2.yr-1 ?

Response:

The carbon loss was 49.9 gC $m^{-2}$ in the form of $CO_2$ during fallow period, we also need to take this into account to estimate NBP, i.e.,

58.8+3.9-49.9=12.8 gC $m^{-2}$ $yr^{-1}$

L 31 : phrase pas supportée par les résultats de cet article. Idem dernière phrase : conclusion n'est pas supportée par les résultats de l'étude : pas de high CUE compare à d'autres cropland et pas forcément du à des effets ferti et irrigation. You cannot do CUE forte plus nfluencée par le fait qu'il y ait 2 cultures sur une année plus que par les ferti et irr.

Response:

We appreciate the reviewer's comments, though we do not understand the language, we tried to use the Google translation and got the following:

"Not supported by the results of this article. Idem last sentence: conclusion is not supported by the results of the study: no high CUE compared to other cropland and not necessarily due to ferti and irrigation effects. You can not do stronger CUE more influenced by the fact that there are 2 crops over a year more than by ferti and irr."

If we understand the reviewer in the right way, the reviewer questioned the conclusion associated with CUE comparisons. We realized such conclusions are not strong, so we decide to keep it in discussion alone rather than in abstract and conclusion sections.

2. Introduction (minors issues):

1) L. 50 : C budget components also includes C exported at harvest and C imported at sowing and through organic fertilisation (Ceschia et al., 2010).

Response:

Will incorporate such knowledge and explanation.

2) Note : Use preferentially bibliographic reference in relation with cropland instead of forest (i.e. Ekblad et al., 2005)

Response:

Will revise.

3) L 65 : add Ceschia et al., 2010 in bibliography list (agro-ecosystem).

Response:

Will revise.

4) Freibauer et al., 2004 is missing in bibliography list.

Response:

Will add this reference in bibliography.

5) L. 81 add Béziat et al., 2009

Response:

Will add.

6) L. 84 add Verma et al., 2015 (suyker)

Response:

We guess the reviewer suggested the following paper that we will incorporate in our references.

"Suyker, A.E., S..B. Verma, G.G. Burba, and T.J. Arkebauer. 2005. Gross primary production and ecosystem respiration of irrigated maize and irrigated soybean during a growing season. Agricultural Forest Meteorology131, 180-190."

7) L 90 : add Ceschia et al., 2010 in bibliography

Response:

Will add.

8) L. 92/93 : "More importantly, there remains no consensus on whether agro-ecosystem is a carbon sink or source" : Depending on management practices (i.e. choice of crops in the rotation, leaving crop residues, including cover crop in the rotation, irrigation practices, : : :), but also climatic year, agroecosystem will be C source or sink. There is no single answer for all agro-ecosystem.

Response:

We agree with the reviewer, we will revise this part to clear our point. The diverse conclusions really motivate us to report our results and add knowledge to $CO_2$ budget of cropland.

9) L. 102 : is it correct to consider 5 m as a very high groundwater ?

Response:

We will revise this to explain the range of groundwater table, rather than comment the table. The most area along the Yellow River has a groundwater table between 0-4 m, if

we look at Fig. 1 in the manuscript. Compared with the 20 m or even deeper groundwater table at other groundwater-fed area, this is indeed a high groundwater table.

3. Materials and methods (majors issues):

1) L. 149 : The annual C budget have to be calculated on a full year, i.e from October 23rd 2010 to October 23rd 2011 (for 365 days).

Response:

We reported a full rotation cycle that does not necessarily follow a whole calendar year (from the date sowing winter wheat, through a whole maize season, to the date of sowing wheat of next season), but we will revise this for an annual $CO_2$ budget estimation.

2) L. 178 : 4 points/10 plant samples in wheat season and 3 plant samples in the maize season : globally, samples are too smalls.

Response:

Such number is indeed small, but these samplings in the main growing season will not serve carbon budget estimations, whereas the biomass samplings at harvest were used for NPP estimate. More importantly, we sampled 200 wheat plants and 5 maize plants at each of the 4 point at harvest (L182), another factor we did not present well is that our field is very homogeneous, all these allow us the confidence of the representativeness of biomass samplings.

3) LAI values are close to PAI values, is it really LAI? Or PAI (plant area index, meaning stem+leaf) ? or GLAI (green leaf area index) ?

Response:

The reviewer is right, this value is measured by LAI-2000, and should be plant area index. We will revise this.

4) Moreover, to compare the 2 methods to calculate NPP, it is necessary to collect the plant samples in the footprint or up from prevailing winds. Those informations are crucial and missing in this section.

Response:

We agree with the reviewer, we indeed collected the samples in the footprint. In addition, all experiments were conducted in the footprint of eddy covariance. A previous work has done the footprint analysis (Lei and Yang, 2010b), based on which we designed our experiment. We will add such explanation in the experiment description section.

5) L. 201 : The equation (1) is not the equation used by Reichstein et al., 2005 (neither Lei and Yang, 2010) as mentioned at L. 197. The equation used in your paper is not the best method, you should use the equation described by Reichstein et al., 2005. (Reichstein et al., 2005) Using this equation allow to better estimate the Rref that is temporally varying in an ecosystem. Rref has to be estimated for consecutive 4-day periods by nonlinear regression using the Lloyd & Taylor (1994) model, fixing all parameters except Rref. By this way, for each point in time, an estimate of Reco can be provided with time-dependent parameters and variables indicated by the symbol t in parentheses contrarily to the equation used in your paper (Reichstein et al., 2005).

Response:

We actually used the method proposed by Reichstein et al. (2005), but adopted the temperature response function of respiration used by Lei and Yang (2010a). Equation (1) is the same as that used in Lei and Yang (2010a) (see equation (3) of their work). We will revise this part to describe the details of the method we used.

6) It is essential that an uncertainty analysis is performed to quantify the uncertainty on the annual NEE (and maybe GPP and ER) values as it was done in Richardson & Hollinger (2007) for instance. Moreover, I would include a footprint analysis and an uncertainty analysis associated to footprint.

Response:

We will perform an uncertainty analysis for the eddy covariance measurements. Following the reviewer's advice, we will use the 'successive days approach' proposed by Hollinger and Richardson (2005) to quantify the eddy covariance based carbon flux. The footprint analysis has been done for this site (Lei and Yang, 2010b), we will mention this in the experimental design section, and let the readers know our experiments were all conducted in the footprint of the eddy covariance.

An analysis using other cropping years would allow a direct comparison of the measurements and modelled data could give more credit to the analysis (see below). L. 206 ": : : the eddy covariance system failed from October 23rd 2010 to April 1st 2011": a comparison could be realised to estimate the uncertainty of the model compared to measured fluxes by eddy covariance and validate the use of the SVR model during this period (6 months) in your site. During the same period of the cropping year (before wheat sowing), the exercise aims at modelling fluxes with SVR model from October to April during bare soil period (for a cropping years with measured data) to compare them to measured fluxes. For example, the exercise is possible by modelling fluxes from October to April 2005-2006 by using 2009-2010 for model training. Then, those results would be compared with measured fluxes from October to April 2005-2006. By this way, you will be able to compare measured fluxes with modelled fluxes and estimate the uncertainties of the model to predict fluxes at this site and during this period.

Response:

For the SVR model, we tested the reliability of the model and the evaluation shows the model performed well. We will take the reviewer's advice to evaluate the model's uncertainty.

L. 515, "we have a total of 466 GPP data samples and 483 ER samples" : is that a total of half-hourly or daily data ? With the term "data samples" we understand half-hourly data that is not sufficient.

Response:

We refer the data samples to number of available days, so the 466 GPP and 483 ER data samples are actually correspond 466 and 483 days' measurements, respectively. We will revise these to explain it more clearly.

7) L. 216 : "Rs" term is not introduced – done in L. 242.

Response:

Will revise.

8) L. 219 : 3 replicated pairs of comparative treatments are not enough to reduce the uncertainty associated with spatial variability.

Response:

We think three replicates are sufficient for our case, because our field are very homogeneous, and the standard deviation of replicates shows relatively low spatial variability (See Fig. 2 for soil respiration measurement, a figure from Zhang et al. (2013)).

[Figure]

**Fig. 4.** Seasonal variation in total soil respiration ($R_S$). Error bars indicate the standard deviation of four replicates (in 2009 and 2010) and three replicates (in 2011). PM, planting maize; PW, planting wheat.

Fig.2 seasonal variations of soil respiration from Zhang et al. (2013)

9) L. 244 : where were realised portable soil respiration system measurements ? Those measurements are they contained in the footprint? This is necessary to be comparable to eddy covariance measurements. If no footprint analysis were realised, specify the direction of prevailing wind. Moreover, provide information concerning the threshold for

considering that the fluxes are representative of the plot and what is the percentage of the data filtered by the footprint analysis.

Response:

All soil respirations were measured in the footprint of eddy covariance

We agree with the reviewer, all the soil respiration, crop samplings and other experiments were conducted in the footprint of eddy covariance. We will add such description in the method section.

10) L. 258 : "NEP (also the inverse of NEE with eddy covariance)" : this definition is false. Net Ecosystem Production (NEP) is annual NEE cumulated. In other word, NEE data are integrated over one year (365 days) to obtain annual NEP (Ceschia et al., 2010).

Response:

Will revise.

11) L. 266 : Replace "grain carbon storage" by "the amount of C exported at harvest". Globally, a detailed description of flux treatment is missing (filtering, percentage of gapfilled data, : : :) but also the authors should use the equation described by Reichstein et al., 2005 to partition the flux. I think that a footprint analyses is missing particularly because two methods are compared and associated to calculate NBP. Also, biomass sampling is too small and non-representative of the plot. That leads to impossibility to compare the two methods (eddy covariance and sampling).

Response:

Will revise the text.

We will describe the treatment of flux data, all other questions have already been addressed in previous paragraphs, please refer to the previous response.

4. Results/Discussion/Conclusion (majors issues) :

1) L. 184 : The crop organs are separated: How the authors seperated the roots ? L. 294/295 : 3.0 % and 2.3 % root for wheat and maize, respectively. Those values seem underestimated. Did the authors compared thoses results ?

Response:

We actually made a trench and took all the roots from the upper 30 cm soil for wheat and 50 cm for maize, we then separated roots from the stem manually. We followed the survey work by Jackson et al. (1996) to determine the depth, because 70% of crop roots are in the upper 30 cm soil. Root sampling always suffer some underestimation, because it is not feasible to get all the roots out of the soil. However, the above-ground biomass generally accounts for 90% of the total biomass for crops (Jackson et al., 1996), the sampling uncertainty associated with root biomass will not change much of the NPP estimates.

2) L. 272 : Replace "a yearly total of 7, 072.18 mol m-2" by a mean value.

Response:

Will revise.

3) L. 276 : Replace "The groundwater table fluctuation well follows irrigation event during winter and spring seasons, while follows precipitation during summer and autumn seasons" by "The groundwater table fluctuations follows well irrigation events during winter and spring seasons, while they follows precipitations during summer and autumn seasons".

Response:

Will revise.

4) Figure 5 : Positives values of daily NEE are not caused by low temperatures as lower temperatures during those periods were never below 10c. It might be a filtering problem. Many tests allow to control flux quality data as described in by Foken et al., 2004; Foken & Wichura, 1996; Göckede et al., 2004. Explain or detail the reason of positive daily value of NEE during the full crop development when temperatures are never under 10C. The whole methodology of flux treatment has to be revised.

Response:

We checked the flux treatment and we found this treatment had no problems. The two evident positive NEE appeared at April 21st and May 8th, respectively, these two days indeed experienced inclement weather, it's for sure that the positive NEE is associated with inclement weather rather than a flux treatment problem. We see pronounced PPFD decline (Fig. 3 (b) in manuscript), please also see Fig. 3 here from the half-hourly measurements.

The reviewer is right, the temperature decline was less pronounced though we can see a slight decline in the half-hourly record. But the positive NEE was indeed associated with inclement weather, and should attribute more to radiation decline, so we will add explanation that attributes the carbon assimilation decline to the radiation decline.

[Figure]

Fig. 3 The decline of Photosynthetically Photon Flux Density (PPFD) and air temperature (Ta) during the inclement weather occurring at April 21st and May 8th, the observation is in half-hourly interval.

L. 302 : "In this section, GPP is presented in negative values: : : from the atmosphere" but in L. 328, the GPP is positive : "the seasonal total NEE, GPP and ER are -437.9, 1078.2 and 640.4 gC m-2 for wheat, and -238.8, 779.7 and 540.8 gC m-2 for maize". GPP + ER = NEE,  -1078.2 + 640.4 = -437.9

Response:

We will keep GPP positive consistently throughout the manuscript. Then we write the equation by

NEE=-GPP+ER.

L. 308 : replace "cropping rotation" by "fallow rotation" and simplify the sentence (L. 308-311) "The fallow periods are the main carbon source periods, especially as well as the period at the start of maize season when the crop is tiny and the high temperature greatly favors respiration.

Response:

Will revise.

L. 312 & 319/321: " inclement weather suppressed crop metabolism rate" There is no visible and drastic climatic variation in Fig. 3. It could be a filtering problem. Realise a thorough assessment of the data.

Response:

The inclement weather was evident, and was well reflected by the PPFD decline in Fig.3(b) in the manuscript, please also see our response to the previous question. The inclement weather indeed caused NEE spike, we also checked our data and confirmed that the data had no problem, therefore we do not think it is a data filtering problem.

L. 322/324 : "When water logging occurred in August and September, both the soil heterotrophic respiration and below-ground autotrophic respiration were suppressed to zero". In this case, ER = 100% meaning that RAB is huge : approx. 5 gC m2/day in at the end of August and beginning of September while before RAB represented only 1 gC m2/day. This example reflects the lack of spatial representativity of the measures between the two methods: Either there is a measurement problem for RH and RS when water logging occurred Or RH and RS are highly overestimated most of the time (by this way, those results would be in line with the results reported by Wang et al., 2015 and Moureaux et al., 2008 but also more consistent with aboveground/belowground biomass measurements cited before. This example also show that the footprint analysis is missing

in this study because the measurements of ER and all its component has to be done in the footprint zone to be comparable to eddy covariance measurements. If you consider that negative values indicate carbon removal from the atmosphere, the NPP should also be negative.

Response:

All experiments were conducted in the footprint.

All calculations strictly follow the commonly used procedure we presented in the manuscript, and the soil respiration measurements methods were consistent before and after water logging, but soil respiration measurement at water logging condition showed zero $CO_2$ efflux.

The RAB gradually changed from approx. 1 gC to 5 gC/m2/day, and fluctuated for a few days. The experiments were all conducted in the footprint zone, and the field is very flat and homogeneous. We have not found similar month-long water logging event in other $CO_2$ cycle researches in upland cropland, indicating that we still lack the partitioning of ecosystem respiration in water logging conditions. Here we speculate that the activity of above ground part should be more active during the water logging, and this may be a survival strategy in water logging conditions. None of the researches suggested by the reviewer have experienced so wet conditions as ours, therefore it is reasonable that we have different results. With our results, we can help further understand such mechanism or at least raise such question as to how plant and ecosystem respiration react to super wet conditions.

We keep all sign consistent now. NPP is always positive representing the carbon storage in plant, and GPP is always positive representing total carbon assimilation through photosynthesis.

In L. 330, NPP values are positives.

Response:

We will keep NPP positive to indicate carbon gain of the ecosystem.

L. 333: the C exported at harvest should be a positive term to indicate C back to the atmosphere as in L. 302 they said that the "GPP is a negative term to indicate C removal from the atmosphere". Whatever the definition of the sign convention chosen, ensure that this convention is respected in the whole document. Moreover, describe explicitly the sink/source effect of each component of the C budget as did Whang et al. (2015); cited in the paper (i.e. L. 65). Also add a table with chosen sign convention for each term as did in the Table 2 in Whang et al. (2015). L. 340: How is calculated the "net carbon loss" ?

Response:

To remove such confusion, we will take the common treatment in environmental and ecological community as follows:

GPP representing the carbon assimilated by photosynthesis will be consistently kept positive throughout the manuscript;

all NPP, NEP and NBP are always positive to indicate the carbon gain in ecosystem;

ER, RH, RAB and RAA are always positive to indicate the carbon loss magnitude from ecosystem;

only NEE can be both positive (indicating carbon gain of the atmosphere) and negative (indicating carbon removal from the atmosphere).

To be brief, only NEE sign is defined by referring to carbon flux associated with atmosphere carbon pool, whereas all others refer to carbon flux associated with the ecosystem.

We will take the reviewer's advice to form a table describing the source/sink effect of each carbon component, if the sign strategy we will take remains confusing.

The carbon loss during fallow period is the ecosystem respiration, i.e., derived from eddy covariance measured NEE of this period, ER=NEE (NEE is positive during the fallow period because there is no plant in the field). We will add such explanation to clear this.

L. 340-342 : There is only one rotation period, not two, but two crops in one cropping year.

Response:

The reviewer is right. We just have one cropping year, here we wanted to describe the fallow periods between two crops. We have two fallow periods in one cropping year, i.e., the period between harvesting wheat and sowing maize, as well as that between harvesting maize and sowing wheat of next season. We will change the 'rotation periods' to 'fallow periods'.

L. 350 : detail "management intensities": straws removal or not, use of organic manure or not,

Response:

Will take the reviewer's advice to detail the management intensities.

L 355 to 357: Further develop other differences: Straw management, organic manure? Strong differences for NEP terms between those several years associated to contrasted climatic years? Differences in exportations (at harvest) terms ?

Response:

We agree with the reviewer, and will revise this part by discussing the details of management.

L. 359 to 361: I recommend further analysis of these differences.

Response:

As the field management is very consistent across years, we will further analyze the different weather conditions between the two periods and see if the weather conditions also contribute to the different carbon sink/source behavior.

L. 361 to 365 : "The water logging event is occasionally reported in upland croplands, Terazawa et al. (1992) and Iwasaki et al. (2010) found water logging cause damage to plants, potentially explaining GPP decline in Dold et al. (2017) and also our study. While our study further implies that water logging diminishes ecosystem respiration even more, therefore reduces overall cropland carbon loss. " so it affects certainly the yield and the carbon budget ! Not discussed.

Response:

We agree with the reviewer, we will discuss how the yield change will impact the carbon budget.

L. 368 to 372 : "Comparing with another study at Luancheng site reporting North China Plain as a carbon source (Wang et al., 2015), we found their estimates of GPP (1051 gC m-2) and ER (692 gC m-2) in wheat season are pretty close to our results (GPP of 1078.2 gC m-2, and ER of 640.4 gC m 2), such resemblance probably attributes to irrigations that prohibit both wheats from experiencing water stress." potentially many other causes : way too much focus on the effect of irrigation.

Response:

We agree with the reviewer on that many other factors can cause such resemblance. But the field managements (tillage type, residue treatment, fertilization, etc) are quite similar in these two sites, and the dominant difference seems field water status which is associated with irrigation type. Our speculation remains reasonable, because these two studies have the same cropping system with both winter wheat and summer maize as main crops, residue treatment, etc., so we would like to keep the detailed comparisons.

L. 372 to 373 : "However, maize of two studies exhibit considerable different carbon fluxes. " compare the components of the C budget with other studies for cropland (i.e. Moureaux et al., 2008).

Response:

We did not compare with the wheat in Moureaux et al. (2008), because we are discussing carbon issues of the same crops (i.e., maize). We have tried our best to incorporate as many references as possible to compare, as a way to reveal the characteristics of our cropland.

L. 387 : "featuring the major difference between these two sites " what are the others terms of management?

Response:

Other managements are cropping types, fertilization, fallow, etc. that are very similar, we will detail these managements.

L. 405 : replace "corn" by "grain"

Response:

Will revise.

L. 412 : "NPP/GPP" not reliable in this study, root biomass estimates are probably very underestimated + too few samples.

Response:

The estimate of NPP/GPP is reliable. Because NPP is estimated by two independent methods that provided similar result in each crop season, indicating that NPP estimates are reasonable. Biomass sampling is related to only one of the two methods for NPP estimations.

L. 432 : add "be" before "subject to environnemental"

Response:

Will revise.

L. 437: "ecosystem respiration is dominated by below-ground and above autotrophic respirations" this difference with the literature reveals a methodological problem in this study (see before for details).

Response:

Our site is different from other studies, especially in groundwater table and field soil moisture status. Our results show how the specific field conditions can alter respiration components. Our experimental design and data treatment are reasonable, please also see our response to previous comments.

L. 449 : 10% for the root ratio compared to only 2.3 and 3% in this study.

Response:

It is not feasible to take all roots out of soil, so it is possible that our root biomass was underestimated. We will follow Moureaux et al. (2008) to assume root accounts for 10% of total biomass, which is also recommend by Jackson et al. (1996).

L. 460 : "because of the sufficient samples" too low according to me.

Response:

Given the homogeneous field and crops, our sampling of 200 wheat plants and 5 maize plants at each of the 4 plots are representative.

L. 469 : not enough samples for estimating NPP !

Response:

Our field and crops are very homogeneous, the biomass samplings at harvest for NPP estimate are representative.

 L. 473 : Calculate uncertainties for each component of NBP. Also NBP = 12.8 gC m-2 yr-1 is not significant because of uncertainties associated to the term.

Response:

We agree with the reviewer.

We will revise and incorporate uncertainty analysis for the different components of NBP, we will use the 'successive days approach' to analyze the uncertainty of eddy covariance based carbon flux.

L. 475 : replace "rotation period" by "fallow period".

Response:

Will revise.

L. 479 to 484 : "NEP are 405.5 and 269.1 gC m-2 for wheat and maize. This cropland has high carbon use efficiency (i.e., the NPP/GPP is 73 % and 72 % for wheat and maize, respectively), which indicates this wheat-maize rotation cropland maintains a relatively higher proportion of assimilated carbon via photosynthesis. The high RH/ER (i.e., 59 % and 54 % for wheat and maize, respectively) implies that soil heterotrophic respiration

dominates ecosystem respiration in this cropland. " I have some doubts concerning those conclusions given the small number of soil respiration measurements. In that state, it is difficult to evaluate the results of this study as signs conventions are not explicitly described. Moreover, the sign convention described in L.302 is opposite to the rest of the document. Whatever the sign convention adopted, please respected it in the whole document. Indeed, it is hard to understand which terms are C sources or C sinks and conclude about the results. C. Technical corrections (non exhaustive) Replace the ";" by "." in lines :

Response:

The cropland is very homogeneous, and our soil respiration experiments show that soil respiration has a relatively low spatial variability, also see my response to previous comments.

For the sign problem, we will keep GPP positive consistently throughout the manuscript. All signs used will be commonly adopted by the community. GPP are positive representing carbon assimilation, ER is positive representing respiration of the ecosystem, whereas only NEE can be either positive or negative.

Will replace ';' with '.'.

294, 378 Add dots in units, i.e. replace "m m-3" by "m.m-3" (L. 274, L. 286). Fig. 3c: irrigation events are difficult to distinguish on the figure. Figure 4: add standard deviations. Figure 6: explain abbreviations HW. And PM.

Response:

We appreciate the reviewer's advice, but we checked lots of published papers, also in Biogeosciences, it is more common to use unit without a dot, so we will not replace them.

The standard deviation will be added.

HW and PM are Harvesting Wheat and Planting Maize, respectively, and we will add this in the figure caption.

References used in this response

Hollinger, D. Y., and Richardson, A. D.: Uncertainty in eddy covariance measurements and its application to physiological models, Tree physiol. 25, 873-885, doi: 10.1093/treephys/25.7.873, 2005.

Jackson, R. B., Canadell, J., Ehleringer, J. R., Mooney, H. A., Sala, O. E., Schulze, E. D.: A global analysis of root distributions for terrestrial biomes, Oecologia, 108, 389-411, doi:10.1007/BF00333714, 1996.

Lei, H. M., and Yang, D. W.: Seasonal and interannual variations in carbon dioxide exchange over a cropland in the North China Plain, Global Change Biol., 16, 2944-2957, doi: 10.1111/j.1365-2486.2009.02136.x, 2010a.

Lei, H. M., and Yang, D. W.: Interannual and seasonal variability in evapotranspiration and energy partitioning over an irrigated cropland in the North China Plain, Agric. For. Meteorol., 150, 581-589, doi: 10.1016/j.agrformet.2010.01.022, 2010b.

Moureaux, C., Debacq, A., Hoyaux, J., Suleau, M., Tourneur, D., Vancutsem, F., Bodson, B., and Aubinet, M.: Carbon balance assessment of a Belgian winter wheat crop (*Triticum aestivum* L.), Global Change Biol., 14, 1353-1366, doi: 10.1111/j.1365-2486.2008.01560.x, 2008.

Reichstein, M., Falge, E., Baldocchi, D., Papale, D., Aubinet, M., Berbigier, P., Bernhofer, C., Buchmann, N., Gilmanov, T., Granier, A., Grunwald, T., Havrankova, K., Ilvesniemi, H., Janous, D., Knohl, A., Laurila, T., Lohila, A., Loustau, D., Matteucci, G., Meyers, T., Miglietta, F., Ourcival, J. M., Pumpanen, J., Rambal, S., Rotenberg, E., Sanz, M., Tenhunen, J., Seufert, G., Vaccari, F., Vesala, T., Yakir, D., and Valentini, R.: On the separation of net ecosystem exchange into assimilation and ecosystem respiration: review and improved algorithm, Global Change Biol., 11, 1424-1439, doi: 10.1111/j.1365-2486.2005.001002.x, 2005.

Suyker, A.E., Verma, S. B., Burba, G. G., and Arkebauer, T. J., Gross primary production and ecosystem respiration of irrigated maize and irrigated soybean during a growing season. Agric. For. Meteorol., 31, 180-190, doi: 10.1016/j.agrformet.2005.05.007, 2005.

Zhang, Q., Lei, H. M., and Yang, D. W.: Seasonal variations in soil respiration, heterotrophic respiration and autotrophic respiration of a wheat and maize rotation cropland in the North China Plain, Agric. For. Meteorol., 180, 34-43, doi: 10.1016/j.agrformet.2013.04.028, 2013.